# Resolving Disagreement Problems in Explainable Artificial Intelligence Through Multi-Criteria Decision Analysis

## Abstract

Post-hoc explanation methods are critical for building trust in complex black-box artificial intelligence (AI) models; however, they often suffer from the disagreement problem, which provides conflicting explanations for the same prediction. This inconsistency undermines reliability and poses a significant barrier to adoption in high-stakes domains that demand trustworthiness and transparency. To address this, we move beyond the search for a single best method and instead propose a principled, preference-driven framework for selecting the best suitable explanation technique for a given context: *which specific post-hoc explanation methods to use and when?* We formalize this selection process as a Multi-Criteria Decision Analysis (MCDA) problem. Our framework evaluates a set of state-of-the-art post-hoc explanation methods (e.g., LIME, BayesLIME, SHAP, BayesSHAP, and Anchor) against six explanation evaluation metrics: fidelity, identity, stability, separability, faithfulness, and consistency. We then apply a suite of established MCDA techniques such as Simple Additive Weighting (SAW), Technique for Order of Preference by Similarity to Ideal Solution (TOPSIS), and Elimination and Choice Translating Reality (ELECTRE I) to aggregate these evaluations based on user-defined priorities. By comparing the rankings produced by these diverse decision logics across multiple predictive models and real-world datasets, we demonstrate not only how to select the optimal explanation method under different priority scenarios (e.g., favoring fidelity vs. stability) but also how to expose critical trade-offs that are invisible to simpler aggregation approaches. Our work provides a robust, transparent, and adaptable methodology for preference-aware explainer selection, transforming explanation disagreement into a structured and justifiable decision-making process.

## 1 Introduction

The deployment of complex machine learning (ML) and deep learning (DL) models in high-stakes domains, such as finance, law, and healthcare, hinges on our ability to trust their decisions Davenport & Kalakota (2019); Cao (2020). Post-hoc explainable AI (XAI) methods, such as LIME Ribeiro et al. (2016), SHAP Lundberg & Lee (2017), etc., are crucial tools for building trust by offering insights into the rationale behind ML/DL model predictions. However, these methods introduce a significant paradox: while designed to enhance the transparency of ML/DL models, they often generate confusion. When applied to the same prediction, different XAI methods frequently highlight different features as most important, leading to what is known as the *disagreement problem* Krishna et al. (2022); Han et al. (2022); Silva et al. (2025); Laberge et al. (2024). Let us consider a healthcare scenario in which a predictive ML/DL model is being deployed to aid doctors in diagnosing diabetic diseases by analyzing a range of patient data (e.g., medical history, genetic information, lifestyle factors, various clinical measurements, etc.,), as shown in Figure 1. In addition, to enhance the transparency and trustworthiness of this ML/DL-driven diagnostic process, state-of-the-art (SOTA) post-hoc explanation methods (e.g., SHAP and LIME) are employed to explain each model prediction to a doctor. These methods generate detailed explanations to elucidate the reasoning behind the ML/DL model's predictions, helping doctors make informed decisions. However, an issue arises when different explanation methods are applied to the same diagnostic task, each focusing on a different set of features to explain the ML/DL model's predictions. For instance, SHAP identifies features such as blood sugar levels, physical activity, etc., that cause diabetes. At the same time, LIME highlights features such as pregnancies, smoking history, etc., that cause diabetes. Such disparities can lead to uncertainty among doctors and undermine the trustworthiness of the AI-based diagnostic system. The

presence of conflicting explanations can lead doctors to doubt the model's recommendations, as they are unsure which explanation to prioritize or trust. This disagreement may lead doctors to rely more on their own clinical expertise, potentially resulting in inconsistent or suboptimal decisions in some instances. Thus, to ensure the trustworthiness of the overall decision-making process, it becomes essential to address these discrepancies among explanation methods. Standardizing or combining these explanations into a coherent, unified narrative could mitigate trust issues and enhance collaborative decision-making among AI systems and professionals across sectors (e.g., healthcare, finance), raising a critical unanswered question: *which explanation method should I trust?*

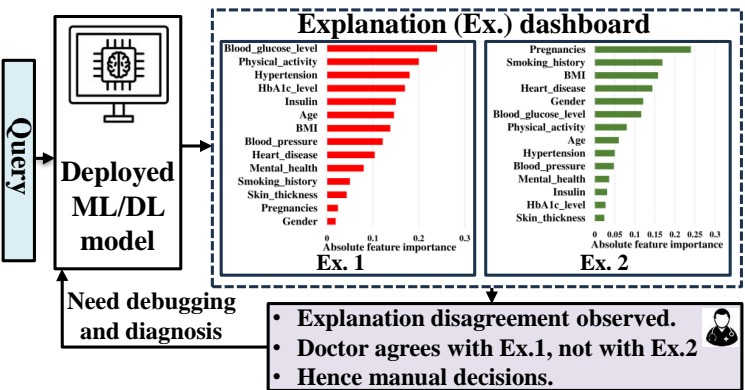

Figure 1: An illustrative example highlights how disagreement in explanation leads to uncertainty among doctors and impacts the trustworthiness of the AI-based diagnostic system.

The root of this disagreement lies in the diverse theoretical foundations of these XAI methods, ranging from game-theoretic Shapley values to local surrogate modeling. To evaluate the quality of these methods, researchers have proposed numerous explanation evaluation metrics assessing desirable properties, such as *fidelity* (how well the explanation reflects the model), *stability* (robustness to minor perturbations) *faithfulness* (if features truly drive the prediction), etc., Alvarez-Melis & Jaakkola (2018); Zhou et al. (2021); Agarwal et al. (2022); Klein et al. (2024). However, this has led to another dilemma: no single XAI method consistently performs well across all metrics. One method may demonstrate high fidelity but poor stability, while another offers greater robustness at the expense of faithfulness. As a result, practitioners face a multi-criteria dilemma, lacking a principled framework to navigate these trade-offs and select the most *suitable explanation method* for their specific context and objectives. The critical research gap, therefore, is not the absence of evaluation metrics but rather the lack of a structured decision-making process to effectively synthesize them.

To address this gap, we formalize the selection of an appropriate explanation method as a *Multi-Criteria Decision Analysis (MCDA)* problem. We propose a novel framework that moves beyond the futile search for a single, universally *"best"* explainer, and instead provides a transparent, preference-driven process for selecting the most appropriate one. Instead of relying on a single aggregation strategy, our core contribution lies in a comparative analysis of three established MCDA techniques with distinct decision-making philosophies: the compensatory Simple Additive Weighting (SAW), the ideal-point-based Technique for Order of Preference by Similarity to Ideal Solution (TOPSIS), and the non-compensatory outranking method ELECTRE I. By applying these techniques, our framework recommends an explanation method that best aligns with user-defined priorities (e.g., favoring fidelity over stability) and, importantly, exposes the inherent trade-offs and sensitivity of the decision. Our key contributions are threefold:

- At first, we empirically show that SOTA post-hoc explanation methods (e.g., LIME, BayesLIME, SHAP, BayesSHAP, and Anchor) exhibit significant performance trade-offs across six key explanation evaluation metrics: fidelity, identity, stability, separability, faithfulness, and consistency, reinforcing the need for a structured decision framework.

- We then propose a novel and adaptable framework that formulates the explanation method selection task as an MCDA problem, enabling user-defined preferences (weights) to guide the decision-making process. Specifically, we apply three MCDA techniques: SAW, TOPSIS, and ELECTRE I to diverse models and datasets, demonstrating how our framework not only provides a suitable explanation method but also uncovers

deeper insights into the decision logic, such as when compensatory vs. non-compensatory reasoning leads to different outcomes.

Evaluated across four ML/DL models, five explanation methods, and three real-world datasets, our approach provides a robust and principled framework for structured, preference-aware explainer selection in a transparent and justifiable manner. For instance, across all datasets, BayesSHAP consistently ranks higher in most scenarios for the predicted class explanations. These findings highlight the often-overlooked issue of explanation disagreement and provide practical guidance for selecting appropriate explainers for predictive models under different evaluation priorities.

## 2 Related Works

In this section, we provide an overview of the current evaluation paradigm in XAI research. Thus, our work builds on the vast literature in XAI. We review prior work and its connections to this research.

SOTA XAI methods have two main approaches: (i) designing inherently interpretable models such as rule lists, decision trees, etc., (ii) applying post-hoc explanation methods, i.e., SHAP, LIME, etc., to explain a black-box ML/DL model locally (for a specific sample) or globally (for the entire model space). Our work focuses on local post-hoc explanation methods based on feature importance. For instance, post-hoc explainers, such as LIME Ribeiro et al. (2016), SHAP Lundberg & Lee (2017), Anchor Ribeiro et al. (2018), BayesLIME Slack et al. (2021), BayesSHAP Slack et al. (2021), etc., are proxy models trained atop a base ML/DL model with the sole intention of *"explaining"* that base model. These methods rely solely on the model's inputs and outputs to identify the salient features that contribute to its output. On the other hand, methods such as SmoothGrad Smilkov et al. (2017), Integrated Gradients Qi et al. (2019), GradCAM Selvaraju et al. (2017), etc., are referred to as gradient-based local explanation methods, which do not use a proxy model but instead compute the gradients of a model with respect to input features to identify important features. A more detailed treatment of this topic is provided in other comprehensive survey articles Madsen et al. (2022); Zhao et al. (2023). However, one problem with these explanation methods is that they often suffer from the *disagreement problem* Krishna et al. (2022); Fazelpour & Fleisher (2025); Silva et al. (2025). From a conceptual standpoint, the misalignment of goals among explanation methods leads to an inconsistent view of explanations Reingold et al. (2024). For instance, the SHAP method is based on game theoretic concepts Lundberg & Lee (2017), whereas LIME is motivated by the function approximation method Ribeiro et al. (2016). Such differences pose conceptual and practical challenges for understanding and using explanation methods, thwarting progress in the XAI field and raising questions about which explanation method to use and when.

Prior research has proposed several metrics to assess the reliability of an explanation. An extensive survey of metrics for evaluating explanation methods is provided in Zhou et al. (2021); Klein et al. (2024); Agarwal et al. (2022). Their survey highlights two high-level goals of explanation methods: interpretability (the clarity, simplicity, and broadness of the explanations) and fidelity (the completeness and soundness of the explanations). In contrast, authors in Liu et al. (2021) proposed a synthetic benchmark for explanation evaluation, including implementations of several metrics and a discussion of how to choose evaluation metrics. Neely et al. Neely et al. (2021) measured the disagreement problem in XAI for the first time, in which they compared different explanation methods (e.g., LIME, DeepSHAP, etc.,) with a rank correlation (Kendall's $\tau$) metric. Instead of only using a rank correlation metric, authors in Krishna et al. (2022) used different evaluation metrics (e.g., feature agreement, rank agreement, sign agreement, etc.,) to capture specific aspects of explanation disagreement w.r.t. the top-k features using various types of data. Unfortunately, their proposed approach did not address how to choose an accurate and trustworthy explanation for the predictive model's decisions. This is important since unstable and inconsistent explanations may lead to an untrustworthy model for end users (e.g., doctors and policymakers). Therefore, Han et.al Han et al. (2022) extend the work of Krishna et al. Krishna et al. (2022) to investigate why the disagreement problem exists for these explanation methods. They conclude that different explanation methods approximate predictive black-box models over different neighborhoods by applying different loss functions. However, one problem with their work is that it focuses on faithfulness rather than on the interpretability of explanations. On the other hand, Avi et.al Schwarzschild et al. (2023) showed the disagreement problem using the Post hoc Explainer Agreement Regularization (PEAR) loss term that measures the difference in feature attribution between a pair of explainers. A more focused disagreement problem study can be found in Roy et al. (2022) where the explanations of LIME and SHAP are investigated for one single defect prediction model. They conclude that LIME and SHAP disagree more on the ranking of important features than on their signs of importance. In addition to these evaluation metrics, human-grounded evaluation approaches emphasize how human

users perceive and utilize explanations Linardatos et al. (2020); Schmitt et al. (2024); Doshi-Velez & Kim (2017); Swamy et al. (2025). For instance, authors in Kaur et al. (2020) found that explanations are often over-trusted and misused. Although our work does not involve human subject studies, we view this line of human-centered investigation as a critical complement to our quantitative evaluation framework. Unfortunately, none of these studies considered how to select an accurate explanation method for the black-box ML/DL model's decisions. This is important because unstable, inconsistent explanations may lead to untrustworthy decisions by end users. Furthermore, the quality of the explanations and the corresponding evaluation metrics were used to analyze the behavior of the explanation methods. Since SOTA explanation methods may result in explanations that are not only inconsistent and unstable but also prone to adversarial attacks and potentially mislead end users into believing that the underlying models and explanations are fair Ghorbani et al. (2019); Slack et al. (2020); Madsen et al. (2022). This, in turn, could lead to the deployment of unfair models in critical real-world applications and ought not to be used in high-stakes decisions. Indeed, it is important to understand and quantify how often explanation methods disagree, and to examine how ML practitioners and domain experts can address such disparities. This is concerning, as inconsistent or misleading explanations can erode trust and propagate unfair decisions in high-stakes settings Madsen et al. (2022).

## 3 Methodology

This section presents the details of our proposed methodology for a trustworthy explanation selection from a set of explanation methods. As illustrated in Figure 2, the proposed method is comprised of three phases: (1) Classification model and explanation method development, (2) Explanation evaluation metrics development, and (3) MCDA method. The details are as follows.

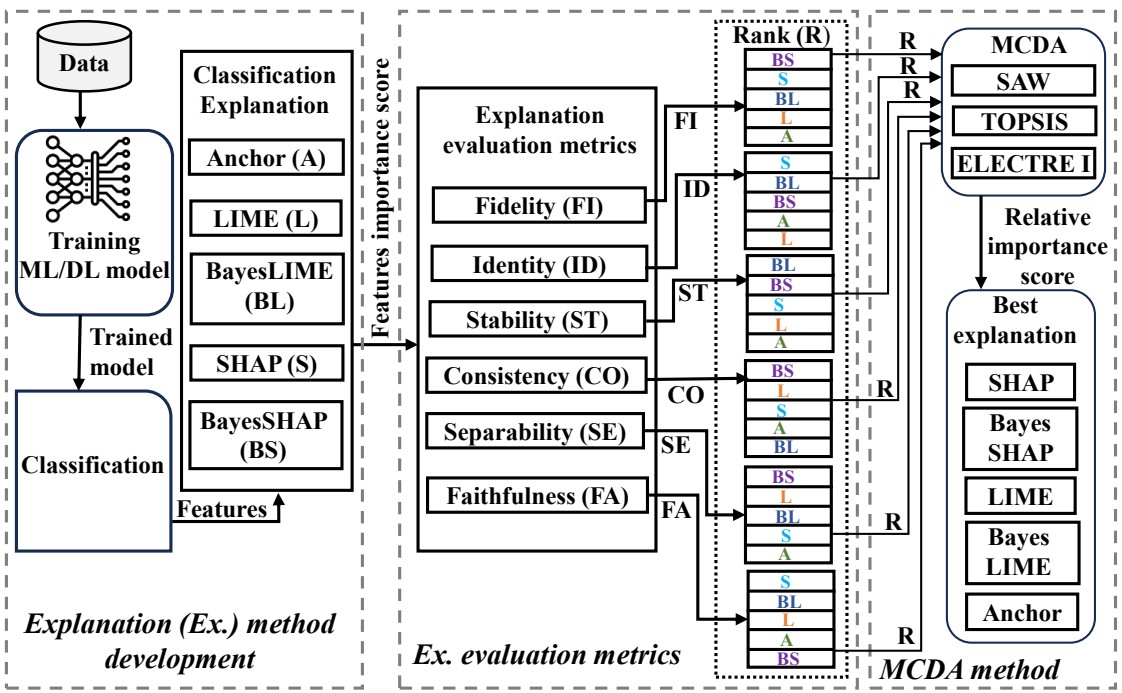

Figure 2: Our proposed framework for selecting an appropriate explanation method from a set of explanation methods.

### 3.1 Explanation Method Development

At first, we develop a classification model using three ML models, namely Random Forest (RF), XGBoost (XGB), Logistic Regression (LR), and a 3-layer neural network (NN) with 50, 100, and 200 nodes per layer, ReLU activation, binary cross-entropy loss, Adam optimizer (learning rate 0.001), and early stopping strategy (patience value = 30) over 100 epochs. These models are trained on three real-world datasets: HELOC Demajo et al. (2020), German Credit Asuncion & Newman (2007), and COMPAS Angwin et al. (2016). Hyperparameter tuning is performed using a grid search, and performance evaluation is conducted through 10-fold cross-validation. Then, to

interpret ML/DL models' predictions, we apply five local post-hoc explanation methods, such as LIME Ribeiro et al. (2016), BayesLIME Slack et al. (2021), SHAP Lundberg & Lee (2017), BayesSHAP Slack et al. (2021), and Anchor Ribeiro et al. (2018)to compute feature importance scores for randomly selected test samples. For BayesLIME and BayesSHAP, the feature-importance scores are computed by running the corresponding Bayesian explainers with 100 perturbations for each test instance, similar to the prior work Slack et al. (2021). Since Anchor produces rule-based explanations rather than direct feature-attribution vectors, we convert each extracted rule into a feature-importance vector by assigning the rule precision to the features included in the rule and zero to all remaining features. It is important to note that this study focuses solely on local explanation methods, where samples are randomly selected from the test dataset. We adopt widely used ML and DL models, as well as five post-hoc explanation techniques and three datasets, aligning with prior works in the XAI disagreement literature Laberge et al. (2024); Han et al. (2022); Krishna et al. (2022).

### 3.2 Explanation Evaluation Metrics

Several metrics have been proposed to evaluate the explainability of XAI methods Alvarez-Melis & Jaakkola (2018); Agarwal et al. (2022). A comprehensive survey by Zhou et al. (2021) identifies two key qualities of high-quality explanations: how well it approximates the model and how human-understandable it is. In this work, we focus on six explanation evaluation metrics: fidelity, identity, stability, separability, consistency, and faithfulness, which help measure the explanation disagreement problem in a principled manner. These metrics are commonly used in recent XAI studies to assess explanation reliability across different contexts Barr et al. (2023); Agarwal et al. (2022); Dai et al. (2022); Bobek et al. (2021); Parimbelli et al. (2023); Solís-Martín et al. (2023); Klein et al. (2024). It is important to note that our goal is not to introduce new explanation evaluation metrics, but rather to evaluate explanation disagreement using established SOTA metrics from prior work Barr et al. (2023); Agarwal et al. (2022); Dai et al. (2022); Bobek et al. (2021); Parimbelli et al. (2023); Solís-Martín et al. (2023); Klein et al. (2024). Accordingly, we adopt similar metric definitions and formulations consistent with the existing literature, ensuring our analysis remains grounded in widely accepted evaluation practice. This aligns with the paper's main goal: not to propose a new family of evaluation metrics, but to provide a structured decision-making framework for synthesizing existing evaluation criteria when no single explainer is uniformly best. The details of these explanation evaluation metrics are provided below.

**Fidelity:** A key question in evaluating explanation quality is *"To what extent does explanation accurately represent the underlying decision-making process?"* Dai et al. (2022). Explanations that correctly identify important features are said to have high fidelity. Note that since true human-annotated feature importance is unavailable, we use *model-derived reference feature-importance scores* from the black-box model $g$ for each input $x$ as the comparison target. Fidelity is quantified as:

$$F_{(x,g,\epsilon)} = \frac{|top(k, \boldsymbol{V}) \cap top(k, v)|}{k}$$

Here, where $v$ denotes the model-derived reference feature-importance scores of $g$ for input $x$, and $\mathbf{V} = \epsilon(x, g)$ is the explanation generated by explanation method $\epsilon$. The fidelity score $F$ therefore measures the fraction of top-$k$ features identified by the explanation that overlap with the top-$k$ features identified from the model-derived reference scores, where $top(k, \cdot)$ returns the indices of the $k$ largest elements. Intuitively, a high-fidelity explanation should recover the features that are most influential according to the model itself. If perturbations to features outside the top-$k$ set produce large changes in the prediction, then the explanation does not faithfully capture the model's decision process. In our experiments, we set $k = 5$ for all top-$k$ sets.

**Stability:** This metric, also known as explanation robustness, captures the idea that similar inputs should yield similar explanations. We use the Lipschitz indicator proposed by Alvarez-Melis and Jaakkola Alvarez-Melis & Jaakkola (2018) to quantify stability, denoted as:

$$S_X(x_i) = \max_{x_j \in N_\varepsilon(x_i)} \frac{||M_i - M_j||}{||x_i - x_j||}$$

Here, $S_X(x_i)$ denotes the stability of test sample $x_i \in X$, with $M_i$ and $M_j$ representing feature importance scores for $x_i$ and its neighbor $x_j$, respectively. The neighborhood $N_\varepsilon(x_i)$ includes points within L2 distance $\epsilon$ of $x_i$. Intuitively,

stability measures how much explanations change under small input perturbations; lower Lipschitz values indicate higher robustness at point $x_i$ Alvarez-Melis & Jaakkola (2018). In this work, the Euclidean distance is used for both numerator and denominator normalization, the neighborhood radius is fixed at $\varepsilon = 0.1$, and neighboring samples are generated from Gaussian noise with $\mathcal{N}(0, 0.05^2)$.

**Faithfulness:** An explanation is faithful if the features it highlights genuinely influence the model's prediction. One way to measure this is through the Prediction Gap on Important features (PGI), which quantifies the change in model output when top-$k$ important features (as identified by an explanation) are perturbed Agarwal et al. (2022). A larger gap implies higher faithfulness. PGI is defined as: $U_{(x,g)} = \frac{1}{p} \sum_{i=1}^{p} |g(x) - g(\tilde{x}_i)|$, where $U_{(x,g)}$ is the faithfulness score for input $x$, $\tilde{x}_i$ is the perturbed version of $x$ (top-$k$ features modified), $g$ is the black-box model, and $p$ is the number of stochastic perturbation runs. Following Barr et al. (2023), we apply Gaussian noise ($\mathcal{N}(0, 0.1)$) for continuous features and flip 30% of discrete ones. Note that, in this study, PGI is used as the only faithfulness metric Agarwal et al. (2022), and we set $p = 10$ perturbation runs per instance.

**Consistency:** This metric assesses how similar the explanations from different methods are for the same prediction Bobek et al. (2021). The main intuition is that repeated explanations for a single observation should produce similar results; otherwise, it suggests instability in the ML/DL model or the explanation method. Consistency is computed by averaging the L1 distance between pairs of explanations, discussed in Bobek et al. (2021). For a given observation, the consistency score is defined as:

$$C(M_{j_1}, M_{j_2}) = \frac{1}{||M_{j_1} - M_{j_2}||_2 + 1}$$

Where $C(M_{j_1}, M_{j_2})$ measures the consistency for the $j^{th}$ observation, and $M_{j_1}$, $M_{j_2}$ the feature importance from two explanation methods. Higher values indicate stronger agreement, whereas lower values indicate greater disagreement. The final consistency score for a given explanation method is computed by averaging its pairwise consistency scores with all other explanation methods. For instance, the consistency score of SHAP is obtained by averaging its pairwise consistency with LIME, BayesSHAP, BayesLIME, and Anchor.

**Identity:** The underlying assumption is that if there are two identical data points(i.e., the actual and predicted classes) their explanations must be identical Solís-Martín et al. (2023). If this condition is not met, then the black-box ML/DL model either predicted the incorrect class or the explanation method produced an explanation that is not identical. The identity metric can be expressed as follows: $\forall_{i,j}(d(x_i, x_j) = 0 \Rightarrow d(\epsilon_i, \epsilon_j) > 0)$. Where $x$ are samples, $d$ is a distance function, and $\epsilon$ is the explanation vector that explains the prediction of each sample.

**Separability:** This metric evaluates whether dissimilar inputs produce dissimilar explanations Parimbelli et al. (2023); Solís-Martín et al. (2023). The underlying assumption is that non-identical samples should not receive identical explanations, since different inputs are expected to differ in the features that contribute positively or negatively to the model's prediction. However, if two samples differ only in features that do not affect the prediction, the model may still produce the same output, and the explanation method may generate very similar explanations. Therefore, in our setting, lower separability values are preferred because the metric is defined as a violation-based score: it measures how often different inputs are assigned identical or overly similar explanations. Formally, the ideal separability property is expressed as: $\forall_{i,j}(d(x_i, x_j) \neq 0 \Rightarrow d(\epsilon_i, \epsilon_j) = 0)$, where $x$ are input samples, $d$ is a distance function, and $\epsilon$ is the explanation vector for each sample. Thus, a low separability score indicates that violations of this property are rare, whereas a high score suggests that the explainer frequently produces similar explanations for distinct inputs.

### 3.3 Proposed Comparative MCDA Framework for XAI Method Selection

To address the challenge of selecting an appropriate explanation method when faced with disagreements, we propose a novel framework based on MCDA. This framework allows for a structured and transparent evaluation of explanation methods against multiple explanation evaluation metrics, with the flexibility to incorporate user-defined preferences through differential weighting. Instead of relying on a single aggregation rule, we implement and compare three established MCDA techniques: Simple Additive Weighting (SAW), Technique for Order of Preference by Similarity to Ideal Solution (TOPSIS), and ELECTRE I.

### 3.3.1 Justification of Methodological Choices

Our framework's components are chosen to ensure a holistic and robust analysis. We selected six evaluation criteria to cover three key dimensions of explanation quality: **Model Alignment** (Fidelity, Faithfulness), **Robustness** (Stability, Identity, Separability), and **Inter-Method Agreement** (Consistency). For the decision logic, we chose three distinct and widely recognized MCDA families to analyze the sensitivity of the rankings to the aggregation philosophy: **SAW** as a simple compensatory baseline, **TOPSIS** as an ideal-point method, and **ELECTRE I** as a non-compensatory outranking method capable of identifying incomparable alternatives and veto effects often missed by other techniques.

### 3.3.2 Formal Preliminaries

Let $A = \{A_1, A_2, \ldots, A_m\}$ be a set of $m$ candidate explanation methods. Let $C = \{C_1, C_2, \ldots, C_n\}$ be a set of $n$ evaluation criteria, where each criterion can be either a "benefit" (higher is better) or a "cost" (lower is better). The performance of each method $A_j \in A$ on each criterion $C_i \in C$ is captured in a raw decision matrix $\mathbf{X}' = [x'_{ij}]$. Given a user's preferences, represented by a weight vector $W = \{w_1, \ldots, w_n\}$ where $w_i$ is the importance of criterion $C_i$ and $\sum_{i=1}^{n} w_i = 1$, our goal is to find a ranking function $\mathcal{R}$ that maps the decision matrix $\mathbf{X}'$ and weights $W$ to a final rank order of the methods in $A$:

$$\mathcal{R}(\mathbf{X}', W) \to \langle A_{(1)}, A_{(2)}, \ldots, A_{(m)} \rangle \tag{1}$$

where $A_{(k)}$ is the method ranked at position $k$. To achieve this, the raw matrix $\mathbf{X}'$ must first be transformed into a unified, normalized decision matrix $\mathbf{X}$ where all criteria are benefits on a common scale. We define this two-step normalization process as follows:

**1. Cost-to-Benefit Transformation.** For each cost criterion $C_i$, we transform its scores $x'_{ij}$ into benefit scores $x''_{ij}$ using:

$$x''_{ij} = \frac{\max_k(x'_{ik}) - x'_{ij}}{\max_k(x'_{ik}) - \min_k(x'_{ik})} \tag{2}$$

Benefit criteria remain unchanged ($x''_{ij} = x'_{ij}$).

**2. Min-Max Normalization.** We then scale all transformed scores $x''_{ij}$ to a final $[0, 1]$ range to produce the normalized decision matrix $\mathbf{X} = [x_{ij}]$:

$$x_{ij} = \frac{x''_{ij} - \min_k(x''_{ik})}{\max_k(x''_{ik}) - \min_k(x''_{ik})} \tag{3}$$

### 3.3.3 Formal Properties of the Framework

A robust selection framework should satisfy desirable formal properties. We define Monotonicity and show that our compensatory methods adhere to it.

**Definition 1 (Monotonicity).** A selection framework is monotonic if improving the performance of an alternative $A_j$ on any single criterion $C_i$, while holding all other scores constant, does not worsen its final rank. Formally, if $S(A_j)$ is the final score for alternative $A_j$, then for any two decision matrices $\mathbf{X}$ and $\hat{\mathbf{X}}$ that are identical except that $\hat{x}_{ij} > x_{ij}$ for some $i, j$, it must hold that $S(A_j)_{\hat{\mathbf{X}}} \geq S(A_j)_{\mathbf{X}}$.

**Proof Sketch for SAW.** The SAW method satisfies Monotonicity. The score for an alternative $A_j$ is $S_j^{\text{SAW}} = \sum_{k=1}^{n} w_k x_{kj}$. If we increase a single score $x_{ij}$ to $\hat{x}_{ij}$, the new score becomes $\hat{S}_j^{\text{SAW}} = (\sum_{k \neq i} w_k x_{kj}) + w_i \hat{x}_{ij}$. Since $w_i \geq 0$ and $\hat{x}_{ij} > x_{ij}$, it follows that $w_i \hat{x}_{ij} \geq w_i x_{ij}$, and therefore $\hat{S}_j^{\text{SAW}} \geq S_j^{\text{SAW}}$. This ensures that an improvement in any criterion never penalizes an alternative.

### 3.3.4 Simple Additive Weighting (SAW)

The SAW method, also known as the Weighted Sum Model (WSM), is a compensatory technique that calculates a total score for each explanation method. The process is detailed in Algorithm 2. The score, denoted as $S_j^{\text{SAW}}$, for method $A_j$ is calculated as:

$$S_j^{\text{SAW}} = \sum_{i=1}^{n} w_i x_{ij} \tag{4}$$

---

**Algorithm 1** Data Preparation and Normalization

---

    **Input:** Datasets ($D_s$), Models ($M_k$), XAI Methods ($A$), Criteria ($C$).
    **Output:** Normalized Decision Matrix $\mathbf{X}$.
1: Compute raw performance matrix $\mathbf{X}' = [x'_{ij}]$ for each $A_j$ on $C_i$.
2: Initialize transformed matrix $\mathbf{X}''$.
3: **for all** criterion $C_i \in C$ **do**
4:     **if** $C_i$ is a cost criterion **then**
5:         Apply cost-to-benefit transformation (Eq. 2) to get column $i$ of $\mathbf{X}''$.
6:     **else**
7:         Copy column $i$ from $\mathbf{X}'$ to $\mathbf{X}''$.
8: Apply min-max normalization (Eq. 3) to $\mathbf{X}''$ to get $\mathbf{X}$.
9: **return** $\mathbf{X}$.

---

**Algorithm 2** Compensatory MCDA Ranking (SAW & TOPSIS)

---

    **Input:** Decision Matrix $\mathbf{X}$, Weight Vector $W$.
    **Output:** SAW Ranking $R^{\text{SAW}}$, TOPSIS Ranking $R^{\text{TOPSIS}}$.
1: *// SAW Calculation*
2: **for all** $A_j \in A$ calculate $S_j^{\text{SAW}} = \sum_{i=1}^{n} w_i x_{ij}$.
3: $R^{\text{SAW}} \leftarrow$ rank alternatives by $S_j^{\text{SAW}}$ (descending).
4: *// TOPSIS Calculation*
5: Construct weighted matrix $\mathbf{V} = [w_i x_{ij}]$.
6: Determine PIS $A^+$ and NIS $A^-$ from $\mathbf{V}$.
7: **for all** $A_j \in A$ calculate distances $D_j^+$ to $A^+$ and $D_j^-$ to $A^-$.
8: **for all** $A_j \in A$ calculate $CC_j^{\text{TOPSIS}} = D_j^- / (D_j^+ + D_j^-)$.
9: $R^{\text{TOPSIS}} \leftarrow$ rank alternatives by $CC_j^{\text{TOPSIS}}$ (descending).
10: **return** $R^{\text{SAW}}, R^{\text{TOPSIS}}$.

---

where $w_i$ is the weight of criterion $C_i$ and $x_{ij}$ is the normalized performance score of method $A_j$ on that criterion. The explanation methods are then ranked based on their $S_j^{\text{SAW}}$ scores in descending order.

### 3.3.5 Technique for Order of Preference by Similarity to Ideal Solution (TOPSIS)

TOPSIS is an ideal-point method that ranks alternatives based on their simultaneous shortest distance from a best-case solution and farthest distance from a worst-case solution, as detailed in Algorithm 2. The key steps are:

**1. Construct the Weighted Normalized Decision Matrix:** Using the normalized decision matrix of scores $x_{ij}$ and weights $w_i$, we construct a weighted matrix where each element $v_{ij}$ is calculated as $v_{ij} = w_i x_{ij}$.

---

**Algorithm 3** Non-Compensatory MCDA (ELECTRE I)

---

    **Input:** Decision Matrix $\mathbf{X}$, Weight Vector $W$, Thresholds $\bar{c}, \bar{d}$.
    **Output:** Outranking Kernel $P^{\text{ELECTRE}}$.
1: Initialize outranking graph $G$.
2: **for all** ordered pairs $(A_k, A_l)$ where $k \neq l$ **do**
3:     Calculate concordance $c(A_k, A_l) = \sum_{i:x_{ik} \geq x_{il}} w_i$.
4:     Calculate discordance $d(A_k, A_l) = \frac{1}{L} \max_{i:x_{il} > x_{ik}} (x_{il} - x_{ik})$.
5:     **if** $c(A_k, A_l) \geq \bar{c}$ AND $d(A_k, A_l) \leq \bar{d}$ **then**
6:         Add directed edge $A_k \rightarrow A_l$ to $G$.
7: $P^{\text{ELECTRE}} \leftarrow$ find nodes in $G$ with in-degree of 0 (the kernel).
8: **return** $P^{\text{ELECTRE}}$.

---

**2. Determine Ideal Solutions:** The Positive Ideal Solution (PIS), denoted $A^+$, is a vector $\{v_1^+, \ldots, v_n^+\}$ where each element $v_i^+$ is the maximum score achieved on criterion $C_i$ across all explanation methods (i.e., $v_i^+ = \max_j v_{ij}$). Conversely, the Negative Ideal Solution (NIS), denoted $A^-$, is a vector $\{v_1^-, \ldots, v_n^-\}$ where each element $v_i^-$ is the minimum score ($v_i^- = \min_j v_{ij}$).

**3. Calculate Separation Measures:** For each explanation method $A_j$, its Euclidean distance from the PIS is calculated as $D_j^+$, and its distance from the NIS is calculated as $D_j^-$:

$$D_j^+ = \sqrt{\sum_{i=1}^n (v_{ij} - v_i^+)^2} \tag{5}$$

$$D_j^- = \sqrt{\sum_{i=1}^n (v_{ij} - v_i^-)^2} \tag{6}$$

**4. Calculate Relative Closeness:** The final score for each explanation method $A_j$, known as the closeness coefficient $CC_j^{\text{TOPSIS}}$, is calculated as:

$$CC_j^{\text{TOPSIS}} = \frac{D_j^-}{D_j^+ + D_j^-} \tag{7}$$

where $0 \le CC_j^{\text{TOPSIS}} \le 1$. Explanation methods are then ranked in descending order of their $CC_j^{\text{TOPSIS}}$ score.

### 3.3.6 ELECTRE I (Elimination and Choice Translating Reality)

ELECTRE I is a non-compensatory outranking method that builds a preference relation between pairs of alternatives, as detailed in Algorithm 3. It determines if an alternative $A_k$ outranks another alternative $A_l$ (denoted $A_k \succeq A_l$) by constructing concordance and discordance indices.

**1. Define Concordance Index:** The concordance index, $c(A_k, A_l)$, measures the strength of evidence supporting the claim that $A_k$ is at least as good as $A_l$. It is the sum of weights for all criteria where $A_k$ performs greater than or equal to $A_l$:

$$c(A_k, A_l) = \sum_{i:x_{ik} \ge x_{il}} w_i \tag{8}$$

where the summation is over the set of indices $i$ for which the score of explanation method $A_k$ ($x_{ik}$) is greater than or equal to the score of method $A_l$ ($x_{il}$).

**2. Define Discordance Index:** The discordance index, $d(A_k, A_l)$, measures the strength of evidence against the claim that $A_k$ outranks $A_l$. It identifies the criterion where $A_l$ most significantly outperforms $A_k$. Let $L$ be the range of the normalized scale (i.e., $L = 1$ for $[0, 1]$).

$$d(A_k, A_l) = \frac{1}{L} \max_{i:x_{il} > x_{ik}} (x_{il} - x_{ik}) \tag{9}$$

If the set of indices where $x_{il} > x_{ik}$ is empty, then $d(A_k, A_l) = 0$.

**3. Determine Outranking Relation:** An alternative $A_k$ is said to outrank $A_l$ ($A_k \succeq A_l$) if and only if its concordance index meets a minimum threshold $\bar{c}$ and its discordance index does not exceed a maximum threshold $\bar{d}$: $c(A_k, A_l) \ge \bar{c}$ AND $d(A_k, A_l) \le \bar{d}$. The result of ELECTRE I is an outranking graph. Further analysis of this graph (e.g., identifying the *kernel*, which contains all non-outranked alternatives) provides a final set of preferred explanation methods. Unlike scoring methods that produce a total ranking, ELECTRE I's primary output is this kernel set. Therefore, in our results table, we report only the members of the kernel, as the method does not provide a relative ranking for the outranked alternatives.

### 3.3.7 Framework Application and Comparative Analysis

The application of our framework, summarized in Algorithms 1, 2, and 3, involves applying these three MCDA methods to the performance scores of the candidate explanation methods. We analyze the rankings produced under various

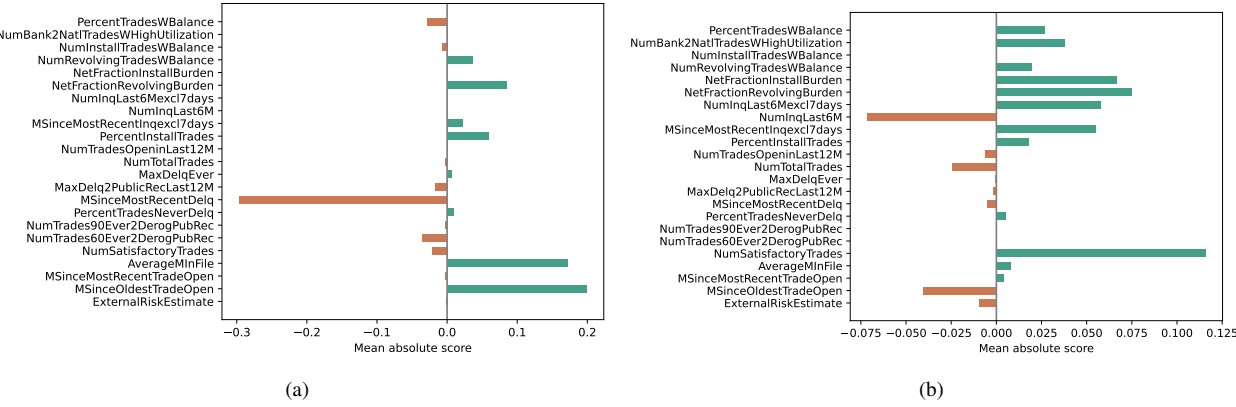

Figure 3: Local explanation using SHAP for a single prediction in the Heloc dataset (**a**) XGB, (**b**) and NN model.

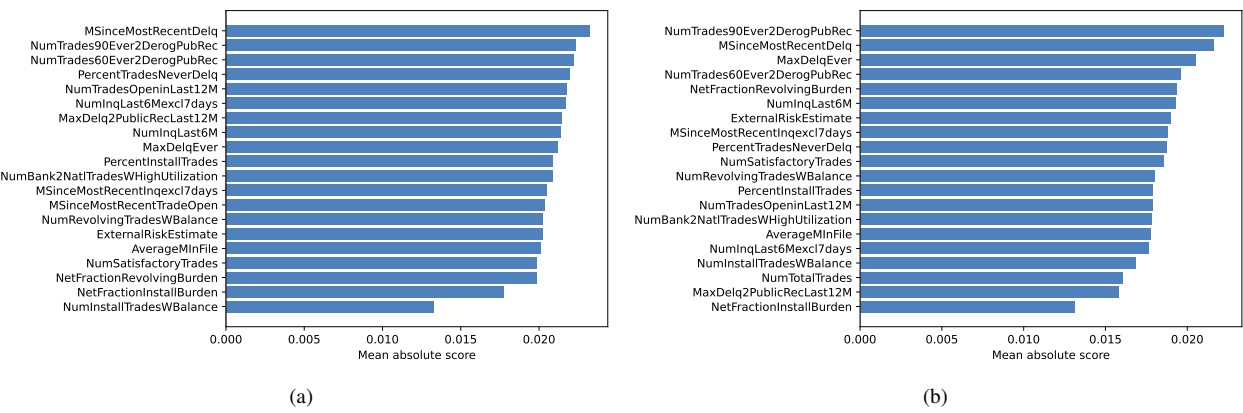

Figure 4: Local explanation using BayeSHAP for a single prediction in the Heloc dataset (**a**) XGB, (**b**) and NN model.

pre-defined weighting scenarios to: (1) identify explanation methods that consistently perform well, (2) understand the sensitivity of the selection to the choice of MCDA technique, and (3) provide practitioners with a transparent tool for selecting an explanation method that best aligns with their requirements.

## 4 Experimental Setup

This section discusses the experimental setup and data we used to evaluate our proposed method. We used Scikit-Learn Pedregosa et al. (2011) to train the ML models and TensorFlow-2.4 Sergeev & Del Balso (2018) to train and evaluate our NN model. To explain the ML/DL models, we used the SHAP Lundberg & Lee (2017), BayesSHAP Slack et al. (2021), Anchor Ribeiro et al. (2018), LIME Ribeiro et al. (2016), and BayesLIME Slack et al. (2021) libraries. All models were trained on an Intel Core i9 Processor and 128GB RAM option with NVIDIA GeForce RTX 3090 Ti GPU.

**Datasets:** To validate the effectiveness of the proposed approach, we use three real-world datasets namely, home equity line of credit (Heloc) Demajo et al. (2020), German Credit Asuncion & Newman (2007), and Correctional Offender Management Profiling for Alternative Sanctions (COMPAS) Angwin et al. (2016). The Heloc dataset is from FICO which consists of information on Heloc applications, with $9,871$ samples for $24$ continuous features. We use this dataset to predict whether an applicant made payments without being 90 days overdue. In contrast, the German Credit dataset comprises of 20 features capturing the bank account balance, loan information, demographics, employment information, and credit history of $1,000$ loan applicants. The class label here is a loan applicant's credit risk (e.g., high or low). However, the COMPAS dataset contains 7 features capturing information about the criminal history, demographics, and prison time of $4,937$ defendants. Each defendant in the dataset is labeled either as high-risk or

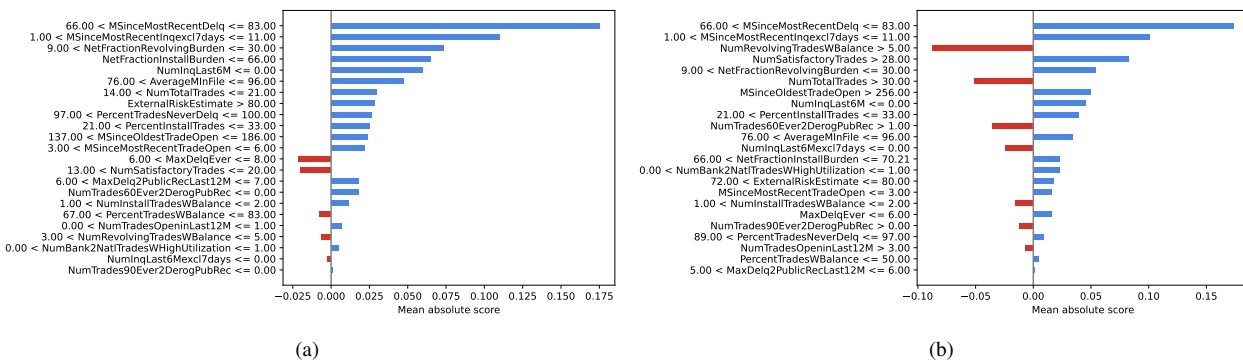

(a)

(b)

Figure 5: Local explanation using LIME for a single prediction in the Heloc dataset, (**a**) XGB, (**b**) and NN model.

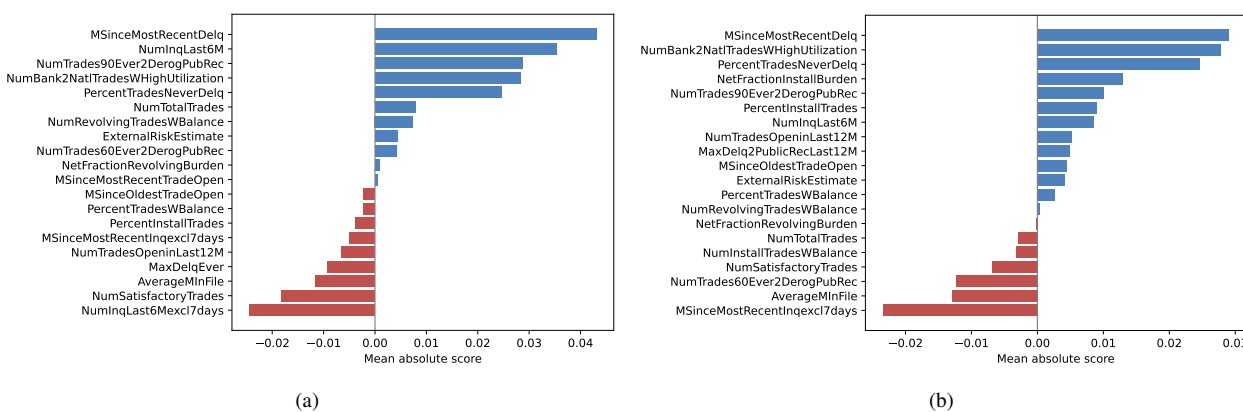

(a)

(b)

Figure 6: Local explanation using BayesLIME for a single prediction in the Heloc dataset, (**a**) XGB, (**b**) and NN model.

low-risk for recidivism based on the COMPAS algorithm's risk score. We used the 70% samples from these datasets to train the ML/DL models and their remaining 30% for testing.

## 5 Experimental Results

This section presents the detailed results of our proposed framework for trustworthy explanation selection from a set of explanation methods.

### 5.1 ML/DL Model Explanation

The local explanation results using the LIME and SHAP methods for the same data instances from the Heloc dataset for the XGB and NN models are shown in Figures 3, 4, 5, 6, and 7. In Figure 5a, the features such as *MSinceMostRecentDel*, *NetFractionRevolvingBurde*, *AverageMInFil*, *ExternalRiskEstimate* etc., are the ones that contribute to bad class classification, while the features such as *MaxDelqEve*, *NumSatisfactoryTrade*, etc., that have a negative effect on bad class classification. In contrast, in Figure 5b, features such as *MSinceMostRecentDel*, *NumSatisfactoryTrade*, *AverageMInFile*,etc., appear to be the most influential features for bad class classification and features such as *NumRevolvingTradesWBalanc*, *NumTotalTrade*, *NumTrades60Ever2DerogPubRec*, etc., are the most dominating features and have a negative impact in bad class classification for the NN model. Interestingly, the SHAP and LIME explanation methods provide different feature rankings for the same instance in the predicted same class (e.g., bad) explanation, leading to disagreement problems. For example, in Figure 3a, *NumSatisfactoryTrades* is the most important feature for bad class classification in the SHAP explanation, while *MSinceMostRecentDelq* is the most important feature for the same class classification in the LIME explanation, as shown in Figure 5a for the NN model. More importantly, we observe disagreement not only across different explanation families, such as SHAP and LIME, but also across closely

---

| IF "MSinceMostRecentInqexcl7day" > 0.59 **AND** "NetFractionRevolvingBurden" <= -7.00 **AND** "PercentTradesNeverDelq"< 97.00 **AND** "ExternalRiskEstimate" > 80 **AND** "NumTotalTrades" <= 1.78 **AND** "PercentTradesWBalance" <= 67.00 **AND** "MaxDelq2PublicRecLast12M"<= 6.00 **AND** "NumTrades90Ever2DerogPubRec"> 0.00 **AND** "PercentInstallTrade" ≤ 32.00 **AND** "NumSatisfactoryTrade" <= 27.00 **AND** "MSinceMostRecentInqexcl7day" <= 11.00 **AND** "MSinceOldestTradeOpen" > 241.00 **THEN Prediction:** "Bad" **WITH** precision = 0.494 **AND** Coverage = 0.062 | IF "PercentInstallTrade" > 0.59 **AND** "PercentTradesWBalance" <= 50.00 **AND** "NumSatisfactoryTrade" > 13.00 **AND** "MSinceMostRecentInqexcl7day" < -1.00 **AND** "NetFractionRevolvingBurden"<=58 **AND** "ExternalRiskEstimate" > 75.00 **AND** "AverageMInFile"<= 96.00 **AND** "NumRevolvingTradesWBalance"<=33.00 **AND** "NumBank2NatlTradesWHighUtilization"<= − 1.00 **AND** "NumInqLast6M"<=0.00 **AND** "NumTotalTrades" <= 30.00 **AND** "MaxDelqEve" <= 6.00 **THEN Prediction:** "Bad" **WITH** precision = 0.678 **AND** Coverage = 0.128 |
|---|---|
| (a) | (b) |

Figure 7: For a single prediction in the Heloc dataset, the local explanations provided by Anchor (**a**) XGB, (**b**) and NN model.

Table 1: Measured explanation evaluation metrics, with average and standard deviation values computed on 50 randomly selected samples from the HELOC dataset. Where ↑ indicates that higher values are better, ↓ indicates that lower values are better, and bolded values are the best values.

| XAI | Model | FI (↑) | ST (↓) | ID (↑) | SE (↓) | CO (↑) | FA (↑) |
|---|---|---|---|---|---|---|---|
| **SHAP** | NN | 1.00 ± 0.00 | 0.39 ± 0.01 | 0.81 ± 0.01 | 0.03 ± 0.01 | 0.40 ± 0.02 | 0.46 ± 0.01 |
| | XGB | 0.96 ± 0.01 | 0.36 ± 0.02 | 0.68 ± 0.02 | 0.03 ± 0.01 | 0.31 ± 0.02 | **0.49** ± 0.01 |
| | RF | 0.94 ± 0.01 | 0.44 ± 0.01 | 0.55 ± 0.02 | 0.05 ± 0.01 | 0.38 ± 0.01 | 0.42 ± 0.01 |
| | LR | 0.91 ± 0.02 | 0.61 ± 0.03 | 0.10 ± 0.03 | 0.21 ± 0.02 | 0.10 ± 0.01 | 0.20 ± 0.01 |
| **BayesSHAP** | NN | **1.00 ± 0.00** | 0.37 ± 0.01 | **0.83 ± 0.01** | **0.02 ± 0.01** | **0.42 ± 0.02** | 0.45 ± 0.01 |
| | XGB | 0.97 ± 0.01 | 0.34 ± 0.01 | 0.71 ± 0.02 | 0.03 ± 0.01 | 0.33 ± 0.02 | 0.48 ± 0.01 |
| | RF | 0.95 ± 0.01 | 0.42 ± 0.01 | 0.58 ± 0.02 | 0.04 ± 0.01 | 0.40 ± 0.01 | 0.40 ± 0.01 |
| | LR | 0.93 ± 0.02 | 0.58 ± 0.02 | 0.13 ± 0.03 | 0.18 ± 0.02 | 0.12 ± 0.01 | 0.24 ± 0.01 |
| **LIME** | NN | 0.99 ± 0.01 | 0.30 ± 0.01 | 0.61 ± 0.02 | 0.03 ± 0.00 | 0.29 ± 0.02 | 0.47 ± 0.02 |
| | XGB | 0.95 ± 0.01 | 0.40 ± 0.01 | 0.67 ± 0.02 | 0.05 ± 0.01 | 0.10 ± 0.01 | 0.35 ± 0.02 |
| | RF | 0.94 ± 0.01 | 0.49 ± 0.02 | 0.28 ± 0.03 | 0.07 ± 0.01 | 0.30 ± 0.02 | 0.29 ± 0.01 |
| | LR | 0.89 ± 0.02 | 0.54 ± 0.02 | 0.08 ± 0.02 | 0.38 ± 0.02 | 0.00 ± 0.00 | 0.19 ± 0.01 |
| **BayesLIME** | NN | 1.00 ± 0.01 | **0.28 ± 0.01** | 0.64 ± 0.02 | **0.02 ± 0.00** | 0.31 ± 0.02 | 0.48 ± 0.02 |
| | XGB | 0.96 ± 0.01 | 0.38 ± 0.01 | 0.69 ± 0.02 | 0.04 ± 0.01 | 0.12 ± 0.01 | 0.37 ± 0.02 |
| | RF | 0.95 ± 0.01 | 0.47 ± 0.01 | 0.31 ± 0.03 | 0.06 ± 0.01 | 0.32 ± 0.02 | 0.31 ± 0.01 |
| | LR | 0.91 ± 0.02 | 0.52 ± 0.02 | 0.10 ± 0.02 | 0.35 ± 0.02 | 0.02 ± 0.00 | 0.21 ± 0.01 |
| **Anchor** | NN | 0.88 ± 0.01 | 0.50 ± 0.01 | 0.10 ± 0.02 | 0.15 ± 0.02 | 0.10 ± 0.02 | 0.30 ± 0.02 |
| | XGB | 0.82 ± 0.01 | 0.50 ± 0.01 | 0.08 ± 0.01 | 0.24 ± 0.02 | 0.09 ± 0.01 | 0.29 ± 0.01 |
| | RF | 0.79 ± 0.02 | 0.56 ± 0.02 | 0.01 ± 0.02 | 0.22 ± 0.01 | 0.10 ± 0.01 | 0.24 ± 0.01 |
| | LR | 0.69 ± 0.02 | 0.64 ± 0.02 | 0.00 ± 0.02 | 0.39 ± 0.02 | 0.00 ± 0.01 | 0.22 ± 0.01 |

related variants, such as SHAP versus BayesSHAP and LIME versus BayesLIME, when explaining the same instance and the same predicted class, as shown in Figures 4 and 6. This suggests that introducing Bayesian uncertainty modeling does not eliminate disagreement but may instead alter how feature importance is distributed and ranked. A similar phenomenon is also observed for the Anchor explanation method, as shown in Figures 7. One thing we notice from these five post-hoc explanation techniques is that they may disagree with each other for the same instance. Indeed, different post-hoc explanation techniques have various goals, leading to an inconsistent and unreliable view of explanation Krishna et al. (2022). Thus, solely looking at the feature ranking does not give sufficient disagreement problems with the predicted class explanation. Therefore, we use six explanation evaluation metrics to measure the intuitions behind the dispute in the predicted class explanation. The next section will evaluate the explanation methods against these metrics.

## 5.2 Performance of Explanation Methods

This section evaluates SHAP, BayesSHAP, LIME, BayesLIME, and Anchor methods across six explanation quality metrics: fidelity (FI), stability (ST), identity (ID), separability (SE), faithfulness (FA), and consistency (CO). To evaluate and measure the disagreement problem in a principled manner, we first randomly select 50 samples from the test datasets. Tables 1, 2, and 3 report the mean and standard deviation of these metrics for the HELOC, German Credit, and COMPAS datasets for the 50 randomly selected test samples. We then extend the analysis to the entire test sets to verify whether the same trends persist under a larger and more representative evaluation protocol, as shown in Tables 4, 5, and 6. Overall, the results from the full test sets remain broadly consistent with those observed on the 50 randomly selected test samples, indicating that the disagreement patterns are not an artifact of small-sample

Table 2: Measured explanation evaluation metrics, with average and standard deviation values computed on 50 randomly selected samples from the German Credit dataset. Where ↑ indicates that higher values are better, ↓ indicates that lower values are better, and bolded are the best values.

| XAI | Model | FI (↑) | ST (↓) | ID (↑) | SE (↓) | CO (↑) | FA (↑) |
|---|---|---|---|---|---|---|---|
| SHAP | NN | 0.99 ± 0.01 | 0.40 ± 0.02 | 0.78 ± 0.02 | 0.05 ± 0.01 | 0.34 ± 0.02 | 0.45 ± 0.02 |
| | XGB | 0.93 ± 0.01 | **0.39 ± 0.01** | 0.69 ± 0.02 | **0.04 ± 0.01** | 0.32 ± 0.01 | 0.43 ± 0.01 |
| | RF | 0.92 ± 0.01 | 0.50 ± 0.02 | 0.48 ± 0.03 | 0.07 ± 0.01 | 0.37 ± 0.02 | 0.32 ± 0.01 |
| | LR | 0.91 ± 0.01 | 0.65 ± 0.02 | 0.01 ± 0.01 | 0.19 ± 0.01 | 0.10 ± 0.01 | 0.15 ± 0.02 |
| BayesSHAP | NN | **1.00 ± 0.01** | 0.38 ± 0.01 | **0.80 ± 0.02** | 0.06 ± 0.01 | 0.36 ± 0.02 | 0.46 ± 0.01 |
| | XGB | 0.95 ± 0.01 | 0.37 ± 0.01 | 0.72 ± 0.02 | **0.04 ± 0.01** | 0.34 ± 0.01 | 0.45 ± 0.01 |
| | RF | 0.94 ± 0.01 | 0.47 ± 0.02 | 0.52 ± 0.02 | 0.06 ± 0.01 | **0.39 ± 0.02** | 0.35 ± 0.01 |
| | LR | 0.92 ± 0.01 | 0.61 ± 0.02 | 0.04 ± 0.01 | 0.17 ± 0.01 | 0.12 ± 0.01 | 0.18 ± 0.01 |
| LIME | NN | 0.97 ± 0.01 | 0.44 ± 0.01 | 0.70 ± 0.02 | 0.05 ± 0.01 | 0.26 ± 0.02 | 0.46 ± 0.01 |
| | XGB | 0.94 ± 0.01 | 0.49 ± 0.02 | 0.75 ± 0.02 | 0.05 ± 0.01 | 0.19 ± 0.02 | 0.39 ± 0.01 |
| | RF | 0.94 ± 0.01 | 0.54 ± 0.02 | 0.40 ± 0.03 | 0.08 ± 0.01 | 0.27 ± 0.02 | 0.31 ± 0.02 |
| | LR | 0.90 ± 0.01 | 0.56 ± 0.02 | 0.10 ± 0.01 | 0.30 ± 0.02 | 0.00 ± 0.01 | 0.15 ± 0.02 |
| BayesLIME | NN | 0.98 ± 0.01 | 0.42 ± 0.01 | 0.72 ± 0.02 | 0.04 ± 0.01 | 0.28 ± 0.02 | **0.49 ± 0.01** |
| | XGB | 0.95 ± 0.01 | 0.46 ± 0.02 | 0.77 ± 0.02 | 0.04 ± 0.01 | 0.21 ± 0.02 | 0.44 ± 0.01 |
| | RF | 0.95 ± 0.01 | 0.51 ± 0.02 | 0.43 ± 0.03 | 0.07 ± 0.01 | 0.29 ± 0.02 | 0.36 ± 0.02 |
| | LR | 0.91 ± 0.01 | 0.53 ± 0.02 | 0.12 ± 0.01 | 0.27 ± 0.02 | 0.02 ± 0.01 | 0.17 ± 0.02 |
| Anchor | NN | 0.84 ± 0.02 | 0.50 ± 0.01 | 0.08 ± 0.01 | 0.17 ± 0.01 | 0.10 ± 0.02 | 0.38 ± 0.02 |
| | XGB | 0.84 ± 0.01 | 0.56 ± 0.02 | 0.01 ± 0.01 | 0.28 ± 0.02 | 0.05 ± 0.01 | 0.25 ± 0.01 |
| | RF | 0.75 ± 0.01 | 0.65 ± 0.02 | 0.00 ± 0.02 | 0.25 ± 0.02 | 0.12 ± 0.01 | 0.32 ± 0.01 |
| | LR | 0.79 ± 0.01 | 0.71 ± 0.02 | 0.00 ± 0.01 | 0.45 ± 0.02 | 0.00 ± 0.01 | 0.29 ± 0.01 |

Table 3: Measured explanation evaluation metrics, with average and standard deviation values computed on 50 randomly selected samples from the Compass dataset. Where ↑ indicates that higher values are better, ↓ indicates that lower values are better, and bolded values are the best values.

| XAI | Model | FI (↑) | ST (↓) | ID (↑) | SE (↓) | CO (↑) | FA (↑) |
|---|---|---|---|---|---|---|---|
| SHAP | NN | 0.95 ± 0.01 | 0.32 ± 0.01 | 0.78 ± 0.01 | **0.04 ± 0.01** | 0.35 ± 0.01 | **0.42 ± 0.01** |
| | XGB | 0.90 ± 0.01 | 0.35 ± 0.01 | 0.71 ± 0.01 | 0.04 ± 0.01 | 0.31 ± 0.01 | 0.41 ± 0.01 |
| | RF | 0.90 ± 0.01 | 0.41 ± 0.02 | 0.41 ± 0.02 | 0.06 ± 0.01 | 0.29 ± 0.01 | 0.37 ± 0.02 |
| | LR | 0.89 ± 0.01 | 0.59 ± 0.02 | 0.10 ± 0.01 | 0.31 ± 0.02 | 0.10 ± 0.01 | 0.27 ± 0.01 |
| BayesSHAP | NN | 0.96 ± 0.01 | **0.27 ± 0.01** | **0.83 ± 0.01** | 0.06 ± 0.01 | **0.39 ± 0.01** | 0.40 ± 0.01 |
| | XGB | 0.92 ± 0.01 | 0.33 ± 0.01 | 0.75 ± 0.01 | 0.05 ± 0.01 | 0.34 ± 0.01 | 0.40 ± 0.01 |
| | RF | 0.91 ± 0.01 | 0.37 ± 0.02 | 0.46 ± 0.02 | 0.08 ± 0.01 | 0.31 ± 0.01 | 0.38 ± 0.02 |
| | LR | 0.90 ± 0.01 | 0.49 ± 0.02 | 0.15 ± 0.01 | 0.34 ± 0.02 | 0.16 ± 0.01 | 0.24 ± 0.01 |
| LIME | NN | 0.96 ± 0.01 | 0.41 ± 0.01 | 0.58 ± 0.02 | 0.05 ± 0.01 | 0.22 ± 0.01 | 0.40 ± 0.01 |
| | XGB | 0.95 ± 0.01 | 0.46 ± 0.02 | 0.49 ± 0.02 | 0.09 ± 0.01 | 0.14 ± 0.01 | 0.33 ± 0.01 |
| | RF | 0.90 ± 0.01 | 0.51 ± 0.02 | 0.35 ± 0.02 | 0.10 ± 0.01 | 0.20 ± 0.01 | 0.30 ± 0.01 |
| | LR | 0.87 ± 0.01 | 0.55 ± 0.02 | 0.01 ± 0.01 | 0.39 ± 0.02 | 0.00 ± 0.01 | 0.15 ± 0.02 |
| BayesLIME | NN | **0.98 ± 0.01** | 0.39 ± 0.01 | 0.61 ± 0.02 | 0.05 ± 0.01 | 0.24 ± 0.01 | 0.41 ± 0.01 |
| | XGB | 0.95 ± 0.01 | 0.43 ± 0.02 | 0.52 ± 0.02 | 0.07 ± 0.01 | 0.16 ± 0.01 | 0.35 ± 0.01 |
| | RF | 0.92 ± 0.01 | 0.48 ± 0.02 | 0.38 ± 0.02 | 0.09 ± 0.01 | 0.22 ± 0.01 | 0.32 ± 0.01 |
| | LR | 0.88 ± 0.01 | 0.52 ± 0.02 | 0.03 ± 0.01 | 0.36 ± 0.02 | 0.02 ± 0.01 | 0.17 ± 0.02 |
| Anchor | NN | 0.82 ± 0.02 | 0.52 ± 0.02 | 0.09 ± 0.01 | 0.21 ± 0.01 | 0.09 ± 0.01 | 0.38 ± 0.02 |
| | XGB | 0.80 ± 0.01 | 0.59 ± 0.02 | 0.01 ± 0.01 | 0.30 ± 0.01 | 0.01 ± 0.01 | 0.29 ± 0.01 |
| | RF | 0.80 ± 0.01 | 0.61 ± 0.02 | 0.00 ± 0.01 | 0.27 ± 0.01 | 0.10 ± 0.01 | 0.31 ± 0.01 |
| | LR | 0.75 ± 0.01 | 0.69 ± 0.02 | 0.00 ± 0.00 | 0.41 ± 0.02 | 0.00 ± 0.01 | 0.25 ± 0.01 |

Table 4: Measured explanation evaluation metrics, with average and standard deviation values computed on entire test samples from the HELOC dataset. Where ↑ indicates that higher values are better, ↓ indicates that lower values are better, and bolded values are the best values.

| XAI | Model | FI (↑) | ST (↓) | ID (↑) | SE (↓) | CO (↑) | FA (↑) |
|---|---|---|---|---|---|---|---|
| SHAP | NN | **1.00 ± 0.01** | 0.44 ± 0.02 | 0.76 ± 0.02 | 0.05 ± 0.01 | 0.36 ± 0.02 | 0.42 ± 0.02 |
| | XGB | 0.99 ± 0.02 | 0.39 ± 0.02 | 0.62 ± 0.02 | 0.08 ± 0.01 | 0.29 ± 0.01 | **0.46 ± 0.02** |
| | RF | 0.96 ± 0.02 | 0.49 ± 0.02 | 0.50 ± 0.03 | 0.12 ± 0.01 | 0.33 ± 0.02 | 0.40 ± 0.02 |
| | LR | 0.91 ± 0.02 | 0.67 ± 0.04 | 0.07 ± 0.03 | 0.29 ± 0.02 | 0.05 ± 0.02 | 0.23 ± 0.02 |
| BayesSHAP | NN | **1.00 ± 0.01** | 0.40 ± 0.02 | **0.80 ± 0.02** | **0.04 ± 0.01** | **0.39 ± 0.01** | 0.43 ± 0.02 |
| | XGB | 0.98 ± 0.02 | 0.38 ± 0.01 | 0.67 ± 0.02 | 0.06 ± 0.01 | 0.31 ± 0.02 | **0.46 ± 0.02** |
| | RF | 0.95 ± 0.02 | 0.44 ± 0.02 | 0.52 ± 0.03 | 0.09 ± 0.01 | 0.34 ± 0.02 | 0.37 ± 0.03 |
| | LR | 0.92 ± 0.02 | 0.62 ± 0.03 | 0.10 ± 0.03 | 0.22 ± 0.02 | 0.10 ± 0.03 | 0.20 ± 0.02 |
| LIME | NN | 0.96 ± 0.02 | 0.36 ± 0.02 | 0.55 ± 0.03 | 0.08 ± 0.01 | 0.25 ± 0.02 | 0.42 ± 0.02 |
| | XGB | 0.93 ± 0.02 | 0.44 ± 0.02 | 0.63 ± 0.03 | 0.09 ± 0.01 | 0.13 ± 0.01 | 0.30 ± 0.02 |
| | RF | 0.90 ± 0.02 | 0.51 ± 0.02 | 0.24 ± 0.03 | 0.13 ± 0.01 | 0.26 ± 0.02 | 0.27 ± 0.02 |
| | LR | 0.86 ± 0.02 | 0.59 ± 0.03 | 0.06 ± 0.02 | 0.38 ± 0.02 | 0.03 ± 0.00 | 0.15 ± 0.02 |
| BayesLIME | NN | 0.99 ± 0.03 | **0.33 ± 0.02** | 0.59 ± 0.03 | 0.06 ± 0.01 | 0.29 ± 0.02 | 0.45 ± 0.02 |
| | XGB | 0.95 ± 0.02 | 0.40 ± 0.02 | 0.65 ± 0.03 | 0.08 ± 0.01 | 0.11 ± 0.01 | 0.35 ± 0.02 |
| | RF | 0.94 ± 0.02 | 0.50 ± 0.02 | 0.27 ± 0.03 | 0.10 ± 0.01 | 0.28 ± 0.02 | 0.26 ± 0.02 |
| | LR | 0.88 ± 0.02 | 0.56 ± 0.03 | 0.09 ± 0.02 | 0.34 ± 0.02 | 0.05 ± 0.01 | 0.17 ± 0.02 |
| Anchor | NN | 0.84 ± 0.02 | 0.52 ± 0.02 | 0.10 ± 0.02 | 0.18 ± 0.01 | 0.12 ± 0.02 | 0.26 ± 0.02 |
| | XGB | 0.79 ± 0.02 | 0.54 ± 0.02 | 0.06 ± 0.02 | 0.28 ± 0.02 | 0.08 ± 0.01 | 0.22 ± 0.02 |
| | RF | 0.76 ± 0.03 | 0.61 ± 0.02 | 0.00 ± 0.04 | 0.25 ± 0.04 | 0.06 ± 0.03 | 0.20 ± 0.02 |
| | LR | 0.67 ± 0.04 | 0.69 ± 0.03 | 0.00 ± 0.03 | 0.47 ± 0.02 | 0.00 ± 0.00 | 0.16 ± 0.03 |

evaluation. It is observed that the BayesSHAP method performs better regarding the NN model most of the time, particularly in terms of fidelity, identity, consistency, and separability. For instance, on the HELOC dataset with 50 randomly selected samples, BayesSHAP, SHAP, and BayesLIME achieve a fidelity score of 1.0 in the NN model, which is 1.02× and 1.14× higher than LIME and Anchor, respectively, indicating relatively low disagreement. A similar trend is preserved when the evaluation is extended to the entire HELOC test set, where SHAP and BayesSHAP again achieve a fidelity score of 1.0, outperforming LIME, BayesLIME, and Anchor, as shown in Table 4. Likewise, on the full COMPAS test set, BayesSHAP remains the best explainer for the NN model, achieving a fidelity score of 0.94, although BayesLIME performs slightly better on the 50 randomly selected COMPAS samples. These results suggest that BayesSHAP consistently provides more robust, agreement-preserving explanations than SHAP, LIME, and BayesLIME when evaluated from a small subset to the full test set. In contrast, Anchor consistently underperforms, particularly in the LR setting, likely due to the mismatch between sparse rule-based explanations and linear decision boundaries Ribeiro et al. (2018). BayesLIME, however, remains highly competitive, especially in terms of stability. For instance, in the HELOC dataset, BayesLIME consistently demonstrates better stability than LIME, SHAP, and BayesSHAP across all models for both the 50 randomly selected samples and the full test set. In addition, the NN model generally exhibits relatively weak stability across all datasets. For instance, in the German Credit dataset, SHAP and BayesSHAP achieve comparatively lower stability scores for the XGB model than for the other ML models in both the 50 randomly selected samples and full-test-set evaluations, indicating reduced robustness in explanation consistency. At the same time, BayesSHAP consistently achieves higher identity scores, indicating stronger consistency across identical inputs in both evaluation settings. For example, on the HELOC dataset with 50 randomly selected samples, BayesSHAP achieves an identity score of 0.83 for the NN model, which is 1.03×, 1.34×, 1.29×, and 7.90× higher than SHAP, LIME, BayesLIME, and Anchor, respectively, highlighting significant disparities in identity and stability across the methods. These findings underscore the need for systematic evaluation when selecting explanation techniques.

Furthermore, for the separability metric, Bayesian variants, particularly BayesLIME and BayesSHAP, generally outperform SHAP and LIME, achieving lower values across most models. For instance, in the HELOC dataset with 50 randomly selected samples, the NN model with BayesLIME and BayesSHAP yields a separability score of 0.02, which is approximately 1.5×, 1.5×, and 5.61× lower than LIME, SHAP, and Anchor, respectively. A similar pattern is also observed on the full test sets of all three datasets, where BayesSHAP, SHAP, and BayesLIME generally remain more separable and less prone to overlap in their explanations than LIME and Anchor, as shown in Tables 4, 5, and 6. Regarding consistency, the values remain far from the ideal score of 1.0, highlighting persistent instability and disagreement across explanation methods. Faithfulness scores are also below ideal. For instance, on the HELOC

Table 5: Measured explanation evaluation metrics, with average and standard deviation values computed on entire test samples from the German Credit dataset. Where ↑ indicates that higher values are better, ↓ indicates that lower values are better, and bolded are the best values.

| XAI | Model | FI (↑) | ST (↓) | ID (↑) | SE (↓) | CO (↑) | FA (↑) |
|---|---|---|---|---|---|---|---|
| SHAP | NN | $0.96 \pm 0.01$ | $0.43 \pm 0.02$ | $0.75 \pm 0.02$ | $0.06 \pm 0.01$ | $0.32 \pm 0.02$ | $0.43 \pm 0.02$ |
| | XGB | $0.94 \pm 0.01$ | $0.40 \pm 0.01$ | $0.67 \pm 0.02$ | $\mathbf{0.05 \pm 0.01}$ | $0.29 \pm 0.01$ | $0.40 \pm 0.01$ |
| | RF | $0.90 \pm 0.02$ | $0.54 \pm 0.02$ | $0.43 \pm 0.03$ | $0.09 \pm 0.02$ | $0.34 \pm 0.02$ | $0.31 \pm 0.00$ |
| | LR | $0.86 \pm 0.01$ | $0.69 \pm 0.02$ | $0.02 \pm 0.01$ | $0.24 \pm 0.01$ | $0.09 \pm 0.01$ | $0.16 \pm 0.02$ |
| BayesSHAP | NN | $\mathbf{0.99 \pm 0.01}$ | $0.39 \pm 0.01$ | $\mathbf{0.81 \pm 0.02}$ | $\mathbf{0.05 \pm 0.01}$ | $0.35 \pm 0.02$ | $0.44 \pm 0.01$ |
| | XGB | $0.95 \pm 0.01$ | $\mathbf{0.38 \pm 0.01}$ | $0.72 \pm 0.02$ | $0.06 \pm 0.01$ | $0.31 \pm 0.01$ | $\mathbf{0.46 \pm 0.01}$ |
| | RF | $0.91 \pm 0.01$ | $0.50 \pm 0.02$ | $0.50 \pm 0.02$ | $0.08 \pm 0.02$ | $\mathbf{0.36 \pm 0.02}$ | $0.31 \pm 0.01$ |
| | LR | $0.90 \pm 0.02$ | $0.63 \pm 0.02$ | $0.05 \pm 0.01$ | $0.19 \pm 0.01$ | $0.11 \pm 0.01$ | $0.16 \pm 0.01$ |
| LIME | NN | $0.93 \pm 0.01$ | $0.47 \pm 0.02$ | $0.68 \pm 0.02$ | $0.07 \pm 0.03$ | $0.24 \pm 0.02$ | $0.43 \pm 0.01$ |
| | XGB | $0.92 \pm 0.01$ | $0.49 \pm 0.02$ | $0.71 \pm 0.02$ | $0.08 \pm 0.01$ | $0.18 \pm 0.02$ | $0.37 \pm 0.01$ |
| | RF | $0.90 \pm 0.01$ | $0.56 \pm 0.02$ | $0.36 \pm 0.03$ | $0.11 \pm 0.01$ | $0.26 \pm 0.02$ | $0.30 \pm 0.02$ |
| | LR | $0.84 \pm 0.01$ | $0.61 \pm 0.02$ | $0.09 \pm 0.02$ | $0.34 \pm 0.02$ | $0.00 \pm 0.01$ | $0.14 \pm 0.02$ |
| BayesLIME | NN | $0.97 \pm 0.01$ | $0.44 \pm 0.01$ | $0.70 \pm 0.02$ | $0.06 \pm 0.01$ | $0.25 \pm 0.02$ | $\mathbf{0.46 \pm 0.01}$ |
| | XGB | $0.94 \pm 0.01$ | $0.48 \pm 0.02$ | $0.75 \pm 0.01$ | $0.07 \pm 0.01$ | $0.20 \pm 0.02$ | $0.40 \pm 0.01$ |
| | RF | $0.92 \pm 0.01$ | $0.54 \pm 0.02$ | $0.40 \pm 0.03$ | $0.10 \pm 0.01$ | $0.26 \pm 0.02$ | $0.33 \pm 0.02$ |
| | LR | $0.87 \pm 0.01$ | $0.57 \pm 0.02$ | $0.10 \pm 0.01$ | $0.30 \pm 0.02$ | $0.01 \pm 0.01$ | $0.17 \pm 0.02$ |
| Anchor | NN | $0.81 \pm 0.02$ | $0.54 \pm 0.02$ | $0.05 \pm 0.01$ | $0.20 \pm 0.01$ | $0.09 \pm 0.02$ | $0.34 \pm 0.02$ |
| | XGB | $0.79 \pm 0.01$ | $0.60 \pm 0.02$ | $0.01 \pm 0.02$ | $0.29 \pm 0.02$ | $0.05 \pm 0.01$ | $0.22 \pm 0.01$ |
| | RF | $0.72 \pm 0.02$ | $0.67 \pm 0.02$ | $0.01 \pm 0.01$ | $0.27 \pm 0.02$ | $0.10 \pm 0.01$ | $0.27 \pm 0.03$ |
| | LR | $0.74 \pm 0.03$ | $0.75 \pm 0.02$ | $0.00 \pm 0.01$ | $0.49 \pm 0.02$ | $0.01 \pm 0.01$ | $0.23 \pm 0.01$ |

Table 6: Measured explanation evaluation metrics, with average and standard deviation values computed on entire test samples from the Compass dataset. Where ↑ indicates that higher values are better, ↓ indicates that lower values are better, and bolded values are the best values.

| XAI | Model | FI (↑) | ST (↓) | ID (↑) | SE (↓) | CO (↑) | FA (↑) |
|---|---|---|---|---|---|---|---|
| SHAP | NN | $0.92 \pm 0.02$ | $0.34 \pm 0.02$ | $0.76 \pm 0.01$ | $\mathbf{0.06 \pm 0.01}$ | $0.33 \pm 0.01$ | $\mathbf{0.40 \pm 0.02}$ |
| | XGB | $0.89 \pm 0.02$ | $0.38 \pm 0.02$ | $0.67 \pm 0.02$ | $0.08 \pm 0.01$ | $0.30 \pm 0.01$ | $0.39 \pm 0.02$ |
| | RF | $0.88 \pm 0.02$ | $0.45 \pm 0.01$ | $0.36 \pm 0.03$ | $0.09 \pm 0.01$ | $0.25 \pm 0.03$ | $0.34 \pm 0.02$ |
| | LR | $0.84 \pm 0.02$ | $0.65 \pm 0.03$ | $0.08 \pm 0.02$ | $0.33 \pm 0.03$ | $0.09 \pm 0.02$ | $0.22 \pm 0.02$ |
| BayesSHAP | NN | $0.94 \pm 0.02$ | $\mathbf{0.30 \pm 0.02}$ | $\mathbf{0.80 \pm 0.02}$ | $\mathbf{0.06 \pm 0.01}$ | $\mathbf{0.36 \pm 0.02}$ | $\mathbf{0.40 \pm 0.02}$ |
| | XGB | $0.90 \pm 0.01$ | $0.36 \pm 0.02$ | $0.71 \pm 0.02$ | $0.07 \pm 0.01$ | $0.31 \pm 0.02$ | $0.37 \pm 0.02$ |
| | RF | $0.90 \pm 0.02$ | $0.41 \pm 0.03$ | $0.42 \pm 0.03$ | $0.10 \pm 0.01$ | $0.30 \pm 0.02$ | $0.33 \pm 0.02$ |
| | LR | $0.87 \pm 0.02$ | $0.52 \pm 0.03$ | $0.13 \pm 0.02$ | $0.39 \pm 0.03$ | $0.11 \pm 0.02$ | $0.20 \pm 0.02$ |
| LIME | NN | $0.93 \pm 0.01$ | $0.43 \pm 0.02$ | $0.54 \pm 0.03$ | $0.08 \pm 0.01$ | $0.19 \pm 0.02$ | $0.35 \pm 0.02$ |
| | XGB | $0.91 \pm 0.02$ | $0.49 \pm 0.03$ | $0.44 \pm 0.03$ | $0.13 \pm 0.01$ | $0.13 \pm 0.02$ | $0.30 \pm 0.02$ |
| | RF | $0.89 \pm 0.02$ | $0.53 \pm 0.03$ | $0.35 \pm 0.03$ | $0.17 \pm 0.01$ | $0.16 \pm 0.02$ | $0.24 \pm 0.02$ |
| | LR | $0.83 \pm 0.02$ | $0.60 \pm 0.03$ | $0.03 \pm 0.01$ | $0.46 \pm 0.03$ | $0.01 \pm 0.01$ | $0.13 \pm 0.02$ |
| BayesLIME | NN | $\mathbf{0.95 \pm 0.02}$ | $0.40 \pm 0.02$ | $0.60 \pm 0.03$ | $0.07 \pm 0.01$ | $0.22 \pm 0.02$ | $0.38 \pm 0.02$ |
| | XGB | $0.92 \pm 0.02$ | $0.47 \pm 0.03$ | $0.48 \pm 0.03$ | $0.10 \pm 0.01$ | $0.15 \pm 0.02$ | $0.30 \pm 0.02$ |
| | RF | $0.90 \pm 0.02$ | $0.50 \pm 0.03$ | $0.35 \pm 0.03$ | $0.12 \pm 0.01$ | $0.18 \pm 0.02$ | $0.29 \pm 0.02$ |
| | LR | $0.86 \pm 0.02$ | $0.56 \pm 0.03$ | $0.02 \pm 0.02$ | $0.39 \pm 0.03$ | $0.01 \pm 0.01$ | $0.14 \pm 0.02$ |
| Anchor | NN | $0.79 \pm 0.03$ | $0.54 \pm 0.03$ | $0.11 \pm 0.02$ | $0.25 \pm 0.02$ | $0.09 \pm 0.02$ | $0.32 \pm 0.03$ |
| | XGB | $0.76 \pm 0.02$ | $0.63 \pm 0.03$ | $0.02 \pm 0.01$ | $0.33 \pm 0.02$ | $0.02 \pm 0.01$ | $0.23 \pm 0.02$ |
| | RF | $0.73 \pm 0.02$ | $0.64 \pm 0.03$ | $0.01 \pm 0.01$ | $0.29 \pm 0.02$ | $0.09 \pm 0.02$ | $0.26 \pm 0.02$ |
| | LR | $0.69 \pm 0.02$ | $0.73 \pm 0.03$ | $0.00 \pm 0.00$ | $0.46 \pm 0.03$ | $0.01 \pm 0.01$ | $0.20 \pm 0.02$ |

Table 7: Comprehensive MCDA rankings of the explanation methods for the NN model, derived from explanation evaluation metric scores computed on 50 randomly selected samples. The table presents the final ranking across three datasets and seven weighting scenarios, highlighting the sensitivity of the selected *best* explanation method to different user priorities. SAW and TOPSIS scores are reported in parentheses (), while for ELECTRE I, the kernel (i.e., the set of non-outranked alternatives) is provided. The abbreviations are as follows: Scenarios (Sc.), framework (Fr.), SHAP (S), BayesSHAP (BS), LIME (L), BayesLIME (BL), Anchor (A), ELECTRE I (ELE I), Balanced (BL), Fidelity-Focused (FI-F), Stability-Focused (ST-F), Identity-Focused (ID-F), Separability-Focused (SE-F), Consistency-Focused (CO-F), and Faithfulness-Focused (FA-F).

| Fr. | Element | HELOC Dataset | | | | | German Credit Dataset | | | | | COMPAS Dataset | | | | |
|-----|---------|------|------|------|------|------|------|------|------|------|------|------|------|------|------|------|
| Sc. | MCDA | R1 | R2 | R3 | R4 | R5 | R1 | R2 | R3 | R4 | R5 | R1 | R2 | R3 | R4 | R5 |
| BL | SAW | BS(0.75) | S(0.71) | BL(0.70) | L(0.68) | A(0.45) | BS(0.68) | S(0.64) | L(0.62) | BL(0.61) | A(0.43) | BS(0.72) | S(0.66) | BL(0.62) | L(0.58) | A(0.48) |
| | TOPSIS | BS(0.91) | S(0.83) | L(0.68) | BL(0.55) | A(0.00) | BS(0.93) | S(0.85) | L(0.75) | BL(0.68) | A(0.00) | BS(0.95) | S(0.89) | BL(0.68) | L(0.57) | A(0.00) |
| | ELE I | {BS} | - | - | - | - | {BS} | - | - | - | - | {BS} | - | - | - | - |
| FI-F | SAW | BS(0.89) | S(0.87) | BL(0.87) | L(0.86) | A(0.79) | BS(0.88) | S(0.86) | BL(0.86) | L(0.85) | A(0.75) | BL(0.85) | BS(0.82) | S(0.81) | L(0.81) | A(0.73) |
| | TOPSIS | BS(1.00) | S(0.97) | L(0.91) | BL(0.85) | A(0.00) | BS(1.00) | S(0.97) | L(0.91) | BL(0.88) | A(0.00) | BL(0.99) | BS(0.92) | L(0.90) | S(0.89) | A(0.00) |
| | ELE I | {BS} | - | - | - | - | {BS} | - | - | - | - | {BL} | - | - | - | - |
| ST-F | SAW | BL(0.76) | L(0.72) | B(0.68) | S(0.66) | A(0.49) | BS(0.67) | S(0.65) | L(0.62) | BL(0.61) | A(0.47) | BS(0.74) | S(0.70) | L(0.61) | BL(0.60) | A(0.50) |
| | TOPSIS | BL(1.00) | L(0.99) | B(0.65) | S(0.61) | A(0.00) | BS(1.00) | S(0.94) | L(0.71) | BL(0.65) | A(0.00) | BS(1.00) | S(0.99) | BL(0.60) | L(0.53) | A(0.00) |
| | ELE I | {BL} | - | - | - | - | {BS} | - | - | - | - | {BS} | - | - | - | - |
| ID-F | SAW | BS(0.82) | S(0.78) | B(0.70) | L(0.68) | A(0.42) | BS(0.79) | S(0.75) | B(0.71) | L(0.69) | A(0.41) | BS(0.81) | S(0.77) | BL(0.63) | L(0.60) | A(0.44) |
| | TOPSIS | BS(1.00) | S(1.00) | L(0.73) | BL(0.50) | A(0.00) | BS(1.00) | S(1.00) | L(0.88) | BL(0.80) | A(0.00) | BS(1.00) | S(1.00) | BL(0.70) | L(0.66) | A(0.00) |
| | ELE I | {BS} | - | - | - | - | {BS} | - | - | - | - | {BS} | - | - | - | - |
| SE-F | SAW | BL(0.80) | BS(0.79) | L(0.78) | S(0.78) | A(0.56) | BL(0.72) | L(0.71) | S(0.71) | BS(0.70) | A(0.53) | S(0.75) | BS(0.74) | BL(0.66) | L(0.64) | A(0.55) |
| | TOPSIS | BL(1.00) | S(0.90) | L(0.90) | BS(0.85) | A(0.00) | BL(1.00) | L(0.83) | S(0.83) | BS(0.79) | A(0.00) | S(0.95) | BS(0.94) | L(0.69) | BL(0.65) | A(0.00) |
| | ELE I | {BL, BS} | - | - | - | - | {BL, S L} | - | - | - | - | {S, BS} | - | - | - | - |
| CO-F | SAW | BS(0.76) | S(0.73) | BL(0.66) | L(0.64) | A(0.43) | BS(0.71) | S(0.69) | BL(0.62) | L(0.59) | A(0.44) | BS(0.75) | S(0.71) | BL(0.59) | L(0.55) | A(0.45) |
| | TOPSIS | BS(1.00) | S(1.00) | L(0.54) | BL(0.45) | A(0.00) | BS(1.00) | S(0.99) | BL(0.75) | L(0.70) | A(0.00) | BS(1.00) | S(1.00) | BL(0.55) | L(0.45) | A(0.00) |
| | ELE I | {BS} | - | - | - | - | {BS} | - | - | - | - | {BS} | - | - | - | - |
| FA-F | SAW | BL(0.72) | L(0.70) | S(0.69) | BS(0.68) | A(0.47) | BL(0.72) | L(0.68) | S(0.67) | BS(0.67) | A(0.49) | S(0.68) | BS(0.66) | L(0.62) | BL(0.62) | A(0.51) |
| | TOPSIS | BL(0.95) | L(0.88) | S(0.83) | BS(0.79) | A(0.00) | BL(1.00) | L(0.95) | S(0.89) | BS(0.87) | A(0.00) | S(0.87) | BS(0.82) | L(0.74) | BL(0.73) | A(0.00) |
| | ELE I | {BL, L} | - | - | - | - | {BL, L} | - | - | - | - | {S, BS} | - | - | - | - |

dataset with 50 randomly selected samples, SHAP achieves the highest mean faithfulness for the XGB model, which is $1.02\times$, $1.40\times$, $1.32\times$, and $1.70\times$ higher than BayesSHAP, LIME, BayesLIME, and Anchor, respectively, indicating that SHAP generally produces more faithful explanations. A similar trend is observed on the full test set. Overall, although the Bayesian variants provide a stronger balance among fidelity, stability, identity, separability, consistency, and faithfulness, **no single explanation method consistently achieves the best faithfulness across all datasets and models**. More importantly, moving from 50 randomly selected samples to the full test sets does not alter the main conclusions of the analysis. Rather, it confirms that the same qualitative trends persist under larger-scale evaluation: BayesSHAP remains the strongest overall explainer in most cases, BayesLIME remains the most competitive alternative to LIME, and Anchor remains the least reliable method. Therefore, the full-test-set results provide stronger evidence of the *disagreement problem*, showing that instability and inconsistency across explanation methods persist even when evaluation is performed on the entire test distribution.

## 5.3 Comparative MCDA Ranking and Analysis

Given our observation that no single explanation method consistently excels across all explanation evaluation metrics, we now apply our proposed comparative MCDA framework to resolve the explanation selection dilemma. The MCDA results for the NN and XGB models based on the 50 randomly selected test samples are presented in Tables 7 and 8, respectively, while the corresponding results for the entire test sets are reported in Tables 9, and 10. Overall, our proposed MCDA analysis reveals a rich and multifaceted performance landscape across three datasets and seven weighting scenarios, and shows that the main ranking patterns remain broadly consistent when the evaluation is extended from a small random subset to the full test set.

First, the results show that the Bayesian variants, particularly BayesSHAP, consistently emerge as robust and well-balanced choices. For example, in the **Balanced** scenario, BayesSHAP ranks first across most datasets and MCDA methods for both the NN and XGB models when using 50 randomly selected samples. Importantly, this trend is preserved in the entire-test-set analysis, where BayesSHAP again remains the dominant or co-dominant alternative in most cases. This dominance extends to priority scenarios focused on explanation integrity: in the **Identity-focused** and **Consistency-focused** scenarios, BayesSHAP is again the clear Rank 1 choice for both 50 randomly selected samples and the entire test set, as shown in Tables 7, 8, 9, and 10. This strong agreement across datasets, scenarios, and decision logics suggests that, for general-purpose use, BayesSHAP provides a well-justified default choice, often offering a modest but consistent improvement over standard SHAP.

Second, our proposed MCDA framework clearly demonstrates that the **optimal explanation method is highly sensitive to both user priorities and the underlying dataset**. Although the SHAP family typically dominates in balanced scenarios, the LIME family becomes much more competitive when a single explanation criterion is emphasized. For instance, in the **Fidelity-focused** scenario for the NN model using the 50 randomly selected samples, SHAP-family methods dominate on HELOC and German Credit, whereas BayesLIME becomes the preferred method on COM-PAS. Similarly, in the **Stability-focused** scenario, BayesLIME and LIME rank among the top methods across several dataset-model combinations, particularly for HELOC under the NN model. These trends remain visible in the full-test-set analysis, as shown in Tables 9, and 10, although the exact scores and rank margins change slightly due to the larger and more representative evaluation pool. These findings empirically invalidate any one-size-fits-all heuristic (e.g., *"always use one type of explainer"*); instead, the preferred explanation method depends strongly on the target metric, the predictive model, and the dataset context, underscoring the need for a data-driven, context-specific evaluation framework like our framework provides.

Third, the comparison among MCDA methods highlights the **important role of decision logic in exposing hidden trade-offs**. In several cases, compensatory methods such as SAW and TOPSIS rank a LIME-family or SHAP-family method first because strong performance on one prioritized criterion compensates for weaker performance on other explanation-quality metrics. For instance, in the **Stability-focused** scenario on the HELOC dataset for the NN model using the 50 randomly selected samples, SAW and TOPSIS rank BayesLIME and LIME first, whereas the non-compensatory ELECTRE I method identifies only BayesLIME as the clear winner, or in some cases retains multiple methods as incomparable alternatives within the kernel. A similar phenomenon is observed in the **Faithfulness-focused** scenarios, where SAW and TOPSIS often favor BayesLIME or LIME, while ELECTRE I either isolates BayesLIME as the only acceptable alternative or retains several methods as non-outranked. The same type of divergence is observed in both the 50-sample and full-test-set analyses, indicating that these trade-offs are structural rather than sample-specific. Therefore, these results show that our proposed MCDA framework not only ranks explanation methods but also reveals whether the final decision is based on smooth compensation among criteria or on a more conservative, risk-aware decision logic.

Finally, across nearly all scenarios, datasets, and model settings, **Anchor is consistently ranked last**, as shown in Tables 7, 8, 9, and 10. This result is observed for both the 50 randomly selected samples and the entire test sets, and holds across SAW, TOPSIS, and ELECTRE I analyses. This suggests that Anchor's sparse rule-based explanation strategy is generally less suitable for these prediction tasks than feature-attribution-based approaches such as SHAP, BayesSHAP, LIME, and BayesLIME. Taken together, the MCDA results reinforce two key conclusions: first, Bayesian explanation variants, especially BayesSHAP, provide the most robust overall performance across multiple scenarios; and second, explanation-method selection remains highly context dependent, which justifies the need for a comparative, multi-criteria framework rather than reliance on any single heuristic explainer choice. Importantly, comparing the 50-sample MCDA results with the entire-test-set MCDA results shows that increasing the evaluation size does not alter the core conclusions of the framework. Instead, the full-test-set analysis confirms the same qualitative ranking trends observed in the smaller-sample setting, while providing stronger evidence that these trends are stable and not an artifact of limited sampling. Thus, the MCDA analysis on the full test sets further strengthens the reliability of the framework for practical explainer selection under varying user priorities.

## 6  Discussion

The experimental results underscore a fundamental shift in how the XAI disagreement problem can be approached. Instead of searching for a single "best" explanation method, our comparative MCDA framework provides a more pragmatic and defensible path: selecting the "most suitable" method for a specific context and set of priorities. This shift carries key implications for practitioners and researchers, which we elaborate on below.

***Implications for Practitioners:*** For ML practitioners and domain experts, this work provides a transparent, structured methodology to navigate the complex landscape of XAI tools. Rather than relying on anecdotal evidence or the popularity of a method, users can articulate their priorities (e.g., *"for this regulatory audit, explanation stability is paramount"*) and receive a well-justified recommendation. The comparison between compensatory (SAW/TOPSIS) and non-compensatory (ELECTRE I) methods is especially valuable, as it highlights the distinction between a *"best on average"* choice and one that avoids critical, "veto-level" weaknesses. For instance, let us recall the healthcare scenario discussed in Section 1, our framework allows a practitioner to evaluate whether SHAP or LIME provides more reliable

Table 8: Comprehensive MCDA rankings of the explanation methods for the XGBoost model, derived from explanation evaluation metric scores computed on 50 randomly selected samples. The table presents the final ranking across three datasets and seven weighting scenarios, highlighting the sensitivity of the selected *best* explanation method to different user priorities. SAW and TOPSIS scores are reported in parentheses (), while for ELECTRE I, the kernel (i.e., the set of non-outranked alternatives) is provided. The abbreviations are as follows: Scenarios (Sc.), framework (Fr.), SHAP (S), BayesSHAP (BS), LIME (L), BayesLIME (BL), Anchor (A), ELECTRE I (ELE I), Balanced (BL), Fidelity-Focused (FI-F), Stability-Focused (ST-F), Identity-Focused (ID-F), Separability-Focused (SE-F), Consistency-Focused (CO-F), and Faithfulness-Focused (FA-F).

| Fr. Element | | HELOC Dataset | | | | | German Credit Dataset | | | | | COMPAS Dataset | | | | |
|---|---|---|---|---|---|---|---|---|---|---|---|---|---|---|---|---|
| Sc. | MCDA | R1 | R2 | R3 | R4 | R5 | R1 | R2 | R3 | R4 | R5 | R1 | R2 | R3 | R4 | R5 |
| BL | SAW | BS (0.70) | S (0.68) | BL (0.62) | L (0.60) | A (0.42) | BS (0.68) | S (0.66) | L (0.62) | BL (0.61) | A (0.39) | BS (0.69) | S (0.66) | L (0.56) | BL (0.55) | A (0.37) |
| | TOPSIS | BS (1.00) | S (1.00) | L (0.78) | BL (0.65) | A (0.00) | BS (0.95) | S (0.92) | L (0.89) | BL (0.80) | A (0.00) | BS (0.98) | S (0.95) | L (0.70) | BL (0.65) | A (0.00) |
| | ELE I | {BS} | - | - | - | - | {BS, S} | - | - | - | - | {BS} | - | - | - | - |
| FI-F | SAW | BS (0.86) | S (0.84) | L (0.82) | BL (0.82) | A (0.70) | BL (0.84) | L (0.83) | BS (0.83) | S (0.82) | A (0.72) | BL (0.81) | L (0.80) | BS (0.80) | S (0.79) | A (0.67) |
| | TOPSIS | BS (1.00) | S (0.97) | L (0.87) | BL (0.85) | A (0.00) | BL (0.98) | L (0.96) | BS (0.93) | S (0.91) | A (0.00) | BL (1.00) | L (0.98) | BS (0.90) | S (0.87) | A (0.00) |
| | ELE I | {BS} | - | - | - | - | {BL, L} | - | - | - | - | {BL, L} | - | - | - | - |
| ST-F | SAW | BS (0.71) | S (0.69) | BL (0.66) | L (0.64) | A (0.46) | BS (0.69) | S (0.67) | L (0.61) | BL (0.60) | A (0.42) | BS (0.71) | S (0.68) | L (0.58) | BL (0.57) | A (0.39) |
| | TOPSIS | BS (1.00) | S (0.95) | L (0.87) | BL (0.80) | A (0.00) | BS (1.00) | S (1.00) | L (0.63) | BL (0.55) | A (0.00) | BS (1.00) | S (1.00) | L (0.62) | BL (0.58) | A (0.00) |
| | ELE I | {BS} | - | - | - | - | {BS, S} | - | - | - | - | {BS, S} | - | - | - | - |
| ID-F | SAW | BS (0.73) | S (0.71) | BL (0.68) | L (0.65) | A (0.39) | BL (0.73) | L (0.70) | S (0.69) | BS (0.69) | A (0.36) | BS (0.74) | S (0.71) | L (0.56) | BL (0.55) | A (0.34) |
| | TOPSIS | BS (1.00) | S (1.00) | L (0.98) | BL (0.80) | A (0.00) | BL (1.00) | L (1.00) | S (0.92) | BS (0.91) | A (0.00) | BS (1.00) | S (1.00) | L (0.61) | BL (0.58) | A (0.00) |
| | ELE I | {BS} | - | - | - | - | {BL, L} | - | - | - | - | {BS, S} | - | - | - | - |
| SE-F | SAW | BS (0.86) | S (0.85) | L (0.80) | BL (0.79) | A (0.64) | BS (0.83) | S (0.82) | L (0.82) | BL (0.81) | A (0.60) | BS (0.84) | S (0.83) | L (0.77) | BL (0.76) | A (0.62) |
| | TOPSIS | BS (1.00) | S (1.00) | L (0.90) | BL (0.85) | A (0.00) | BS (1.00) | S (0.98) | L (0.97) | BL (0.95) | A (0.00) | BS (1.00) | S (1.00) | L (0.88) | BL (0.85) | A (0.00) |
| | ELE I | {BS, S} | - | - | - | - | {BS, S, L, BL} | - | - | - | - | {BS, S} | - | - | - | - |
| CO-F | SAW | BS (0.74) | S (0.72) | BL (0.58) | L (0.56) | A (0.41) | BS (0.70) | S (0.69) | BL (0.60) | L (0.58) | A (0.36) | BS (0.71) | S (0.70) | L (0.53) | BL (0.53) | A (0.33) |
| | TOPSIS | BS (1.00) | S (1.00) | BL (0.20) | L (0.13) | A (0.00) | BS (1.00) | S (1.00) | BL (0.55) | L (0.45) | A (0.00) | BS (1.00) | S (1.00) | L (0.34) | BL (0.33) | A (0.00) |
| | ELE I | {BS, S} | - | - | - | - | {BS, S} | - | - | - | - | {BS, S} | - | - | - | - |
| FA-F | SAW | BS (0.72) | S (0.71) | BL (0.63) | L (0.61) | A (0.45) | BS (0.69) | S (0.68) | L (0.62) | BL (0.62) | A (0.41) | BS (0.69) | S (0.68) | L (0.58) | BL (0.57) | A (0.39) |
| | TOPSIS | BS (1.00) | S (1.00) | BL (0.55) | L (0.47) | A (0.00) | BS (0.95) | S (0.93) | L (0.82) | BL (0.79) | A (0.00) | BS (1.00) | S (0.98) | L (0.76) | BL (0.75) | A (0.00) |
| | ELE I | {BS, S} | - | - | - | - | {BS, S} | - | - | - | - | {BS, S} | - | - | - | - |

Table 9: Comprehensive MCDA rankings of the explanation methods for the NN model, derived from explanation evaluation metric scores computed on the entire test set. The table presents the final ranking across three datasets and seven weighting scenarios, highlighting the sensitivity of the selected *best* explanation method to different user priorities. SAW and TOPSIS scores are reported in parentheses (), while for ELECTRE I, the kernel (i.e., the set of non-outranked alternatives) is provided. The abbreviations are as follows: Scenarios (Sc.), framework (Fr.), SHAP (S), BayesSHAP (BS), LIME (L), BayesLIME (BL), Anchor (A), ELECTRE I (ELE I), Balanced (BL), Fidelity-Focused (FI-F), Stability-Focused (ST-F), Identity-Focused (ID-F), Separability-Focused (SE-F), Consistency-Focused (CO-F), and Faithfulness-Focused (FA-F).

| Fr. Element | | HELOC Dataset | | | | | German Credit Dataset | | | | | COMPAS Dataset | | | | |
|---|---|---|---|---|---|---|---|---|---|---|---|---|---|---|---|---|
| Sc. | MCDA | R1 | R2 | R3 | R4 | R5 | R1 | R2 | R3 | R4 | R5 | R1 | R2 | R3 | R4 | R5 |
| BL | SAW | BS(0.73) | S(0.69) | BL(0.65) | L(0.62) | A(0.41) | BS(0.70) | S(0.66) | BL(0.61) | L(0.58) | A(0.40) | BS(0.72) | S(0.68) | BL(0.60) | L(0.56) | A(0.39) |
| | TOPSIS | BS(0.90) | S(0.82) | BL(0.63) | L(0.50) | A(0.00) | BS(0.92) | S(0.83) | BL(0.65) | L(0.56) | A(0.00) | BS(0.95) | S(0.87) | BL(0.58) | L(0.48) | A(0.00) |
| | ELE I | {BS} | - | - | - | - | {BS} | - | - | - | - | {BS} | - | - | - | - |
| FI-F | SAW | BS(0.88) | S(0.86) | BL(0.84) | L(0.80) | A(0.71) | BS(0.87) | S(0.84) | BL(0.83) | L(0.80) | A(0.69) | BL(0.82) | BS(0.81) | S(0.79) | L(0.78) | A(0.67) |
| | TOPSIS | BS(1.00) | S(0.95) | BL(0.84) | L(0.72) | A(0.00) | BS(1.00) | S(0.94) | BL(0.88) | L(0.79) | A(0.00) | BL(0.98) | BS(0.91) | S(0.86) | L(0.85) | A(0.00) |
| | ELE I | {BS} | - | - | - | - | {BS} | - | - | - | - | {BL} | - | - | - | - |
| ST-F | SAW | BL(0.74) | L(0.71) | BS(0.67) | S(0.64) | A(0.46) | BS(0.70) | S(0.65) | BL(0.61) | L(0.58) | A(0.43) | BS(0.73) | S(0.69) | BL(0.59) | L(0.55) | A(0.41) |
| | TOPSIS | BL(1.00) | L(0.92) | BS(0.66) | S(0.52) | A(0.00) | BS(1.00) | S(0.82) | BL(0.70) | L(0.57) | A(0.00) | BS(1.00) | S(0.90) | BL(0.50) | L(0.40) | A(0.00) |
| | ELE I | {BL} | - | - | - | - | {BS} | - | - | - | - | {BS} | - | - | - | - |
| ID-F | SAW | BS(0.80) | S(0.76) | BL(0.68) | L(0.64) | A(0.38) | BS(0.80) | S(0.75) | BL(0.69) | L(0.66) | A(0.36) | BS(0.79) | S(0.75) | BL(0.60) | L(0.54) | A(0.34) |
| | TOPSIS | BS(1.00) | S(0.95) | BL(0.60) | L(0.45) | A(0.00) | BS(1.00) | S(0.93) | BL(0.77) | L(0.69) | A(0.00) | BS(1.00) | S(0.94) | BL(0.57) | L(0.45) | A(0.00) |
| | ELE I | {BS} | - | - | - | - | {BS} | - | - | - | - | {BS} | - | - | - | - |
| SE-F | SAW | BS(0.84) | S(0.80) | BL(0.74) | L(0.66) | A(0.49) | BS(0.82) | S(0.76) | BL(0.75) | L(0.70) | A(0.52) | BS(0.81) | S(0.80) | BL(0.69) | L(0.64) | A(0.51) |
| | TOPSIS | BS(1.00) | S(0.86) | BL(0.70) | L(0.52) | A(0.00) | BS(1.00) | S(0.86) | BL(0.84) | L(0.76) | A(0.00) | BS(1.00) | S(0.98) | BL(0.78) | L(0.72) | A(0.00) |
| | ELE I | {BS} | - | - | - | - | {BS} | - | - | - | - | {BS,S} | - | - | - | - |
| CO-F | SAW | BS(0.75) | S(0.71) | BL(0.63) | L(0.59) | A(0.40) | BS(0.71) | S(0.67) | BL(0.60) | L(0.56) | A(0.38) | BS(0.74) | S(0.70) | BL(0.56) | L(0.50) | A(0.36) |
| | TOPSIS | BS(1.00) | S(0.88) | BL(0.50) | L(0.32) | A(0.00) | BS(1.00) | S(0.86) | BL(0.48) | L(0.40) | A(0.00) | BS(1.00) | S(0.88) | BL(0.42) | L(0.28) | A(0.00) |
| | ELE I | {BS} | - | - | - | - | {BS} | - | - | - | - | {BS} | - | - | - | - |
| FA-F | SAW | BL(0.72) | BS(0.70) | S(0.66) | L(0.64) | A(0.42) | BL(0.71) | BS(0.69) | S(0.66) | L(0.63) | A(0.45) | BS(0.69) | S(0.67) | BL(0.61) | L(0.56) | A(0.46) |
| | TOPSIS | BL(0.95) | BS(0.88) | S(0.74) | L(0.70) | A(0.00) | BL(0.96) | BS(0.89) | S(0.75) | L(0.68) | A(0.00) | BS(0.92) | S(0.87) | BL(0.65) | L(0.53) | A(0.00) |
| | ELE I | {BL} | - | - | - | - | {BL} | - | - | - | - | {BS,S} | - | - | - | - |

insights for clinicians based on specific diagnostic priorities. By formalizing this selection, our framework helps ensure the chosen explanations are consistent and trustworthy, fostering collaborative decision-making between the AI system and healthcare professionals and mitigating trust issues.

***Methodological Implications for Researchers:*** Our work demonstrates the value of applying established decision science principles to challenges in XAI. The discordance observed between compensatory (e.g., SAW, TOPSIS) and non-compensatory (ELECTRE I) methods highlights that the choice of aggregation logic itself is a crucial, often overlooked, aspect of meta-evaluations in AI. It suggests that future XAI evaluation benchmarks should consider offering a suite of aggregation techniques rather than a single, fixed leaderboard score. Furthermore, for researchers developing novel explanation methods, our framework provides a multi-faceted benchmark. Instead of claiming superiority on a single metric, new methods can be evaluated on their performance profiles across various weighting scenarios, offering a more complete and honest assessment of their strengths and trade-offs against existing techniques. This provides a

Table 10: Comprehensive MCDA rankings of the explanation methods for the XGBoost model, derived from explanation evaluation metric scores computed on the entire test set. The table presents the final ranking across three datasets and seven weighting scenarios, highlighting the sensitivity of the selected *best* explanation method to different user priorities. SAW and TOPSIS scores are reported in parentheses (), while for ELECTRE I, the kernel (i.e., the set of non-outranked alternatives) is provided. The abbreviations are as follows: Scenarios (Sc.), framework (Fr.), SHAP (S), BayesSHAP (BS), LIME (L), BayesLIME (BL), Anchor (A), ELECTRE I (ELE I), Balanced (BL), Fidelity-Focused (FI-F), Stability-Focused (ST-F), Identity-Focused (ID-F), Separability-Focused (SE-F), Consistency-Focused (CO-F), and Faithfulness-Focused (FA-F).

| Fr. Element | | HELOC Dataset | | | | | German Credit Dataset | | | | | COMPAS Dataset | | | | |
|---|---|---|---|---|---|---|---|---|---|---|---|---|---|---|---|---|
| Sc. | MCDA | R1 | R2 | R3 | R4 | R5 | R1 | R2 | R3 | R4 | R5 | R1 | R2 | R3 | R4 | R5 |
| BL | SAW | BS(0.72) | S(0.69) | BL(0.61) | L(0.57) | A(0.39) | BS(0.71) | S(0.66) | BL(0.62) | L(0.58) | A(0.38) | BS(0.70) | S(0.67) | BL(0.60) | L(0.56) | A(0.36) |
| | TOPSIS | BS(0.97) | S(0.95) | BL(0.61) | L(0.46) | A(0.04) | BS(1.00) | S(0.90) | BL(0.73) | L(0.61) | A(0.00) | BS(1.00) | S(0.94) | BL(0.59) | L(0.48) | A(0.00) |
| | ELE I | {BS} | - | - | - | - | {BS,S} | - | - | - | - | {BS} | - | - | - | - |
| FI-F | SAW | BS(0.84) | S(0.82) | BL(0.79) | L(0.75) | A(0.66) | BS(0.84) | S(0.80) | BL(0.79) | L(0.74) | A(0.67) | BL(0.81) | BS(0.79) | L(0.76) | S(0.74) | A(0.63) |
| | TOPSIS | BS(1.00) | S(0.90) | BL(0.78) | L(0.63) | A(0.01) | BS(0.99) | S(0.86) | BL(0.84) | L(0.70) | A(0.00) | BL(1.00) | BS(0.90) | L(0.82) | S(0.76) | A(0.00) |
| | ELE I | {BS} | - | - | - | - | {BS} | - | - | - | - | {BL} | - | - | - | - |
| ST-F | SAW | BS(0.73) | S(0.69) | BL(0.63) | L(0.58) | A(0.40) | BS(0.70) | S(0.66) | BL(0.58) | L(0.54) | A(0.38) | BS(0.72) | S(0.68) | BL(0.57) | L(0.53) | A(0.37) |
| | TOPSIS | BS(1.00) | S(0.88) | BL(0.57) | L(0.38) | A(0.00) | BS(1.00) | S(0.91) | BL(0.45) | L(0.36) | A(0.00) | BS(1.00) | S(0.94) | BL(0.43) | L(0.34) | A(0.00) |
| | ELE I | {BS} | - | - | - | - | {BS} | - | - | - | - | {BS} | - | - | - | - |
| ID-F | SAW | BS(0.75) | S(0.71) | BL(0.66) | L(0.63) | A(0.36) | BL(0.75) | BS(0.72) | L(0.69) | S(0.65) | A(0.34) | BS(0.73) | S(0.69) | BL(0.55) | L(0.51) | A(0.31) |
| | TOPSIS | BS(1.00) | S(0.92) | BL(0.70) | L(0.62) | A(0.00) | BL(1.00) | BS(0.84) | L(0.78) | S(0.66) | A(0.00) | BS(1.00) | S(0.90) | BL(0.46) | L(0.38) | A(0.00) |
| | ELE I | {BS} | - | - | - | - | {BL} | - | - | - | - | {BS} | - | - | - | - |
| SE-F | SAW | BS(0.84) | S(0.80) | BL(0.74) | L(0.69) | A(0.57) | BS(0.82) | S(0.74) | BL(0.78) | L(0.68) | A(0.56) | BS(0.81) | S(0.79) | BL(0.66) | L(0.61) | A(0.52) |
| | TOPSIS | BS(1.00) | S(0.90) | BL(0.78) | L(0.66) | A(0.00) | BS(1.00) | S(0.86) | BL(0.88) | L(0.65) | A(0.00) | BS(1.00) | S(0.95) | BL(0.68) | L(0.55) | A(0.00) |
| | ELE I | {BS} | - | - | - | - | {BS,BL,S} | - | - | - | - | {BS,S} | - | - | - | - |
| CO-F | SAW | BS(0.74) | S(0.70) | BL(0.53) | L(0.50) | A(0.35) | BS(0.71) | S(0.67) | BL(0.55) | L(0.51) | A(0.33) | BS(0.72) | S(0.69) | BL(0.49) | L(0.45) | A(0.30) |
| | TOPSIS | BS(1.00) | S(0.90) | BL(0.30) | L(0.24) | A(0.00) | BS(1.00) | S(0.88) | BL(0.38) | L(0.30) | A(0.00) | BS(1.00) | S(0.92) | BL(0.28) | L(0.20) | A(0.00) |
| | ELE I | {BS} | - | - | - | - | {BS} | - | - | - | - | {BS} | - | - | - | - |
| FA-F | SAW | BS(0.72) | S(0.71) | BL(0.60) | L(0.56) | A(0.40) | BS(0.70) | S(0.67) | BL(0.61) | L(0.57) | A(0.39) | S(0.69) | BS(0.67) | BL(0.55) | L(0.52) | A(0.37) |
| | TOPSIS | BS(1.00) | S(0.98) | BL(0.52) | L(0.42) | A(0.00) | BS(1.00) | S(0.88) | BL(0.58) | L(0.45) | A(0.00) | S(1.00) | BS(0.90) | BL(0.46) | L(0.38) | A(0.00) |
| | ELE I | {BS,S} | - | - | - | - | {BS} | - | - | - | - | {S,BS} | - | - | - | - |

structured approach to identify where a new method is susceptible to disagreement and how it compares to established alternatives.

# 7 Limitations and Future Work

Although our proposed method addresses a key challenge for the practical adoption of XAI, it has a few limitations. First, we evaluated our framework using three ML models and one relatively simple DL model. With the growing interest in large foundation models and generative AI, we have not yet applied our framework to more complex DL architectures, such as Transformer-based models underlying large language models and reasoning systems, as well as diffusion models. Extending our evaluation to these models is an important direction for future work. Second, our study considered five post-hoc explanation methods and six explanation evaluation metrics. While this set is representative, it is not exhaustive and may not capture all relevant aspects of explanation quality. Third, we focused solely on feature-importance-based explanation methods in this work. As part of our future research, we plan to evaluate our proposed method alongside other explanation families, including example-based, counterfactual, and visual methods. Moreover, the weighting scenarios used in our experiments were pre-defined for illustrative purposes. Real-world deployments would benefit from a more formal preference elicitation process involving stakeholders, such as the Analytic Hierarchy Process (AHP) Saaty (2008). While the framework provides a principled and transparent mechanism for ranking explainers, the current manuscript does not yet establish whether the selected explainer improves downstream outcomes such as debugging, trust calibration, error detection, or decision quality. Our goal in this work is to establish the soundness of the decision-analytic framework itself, making the current study a methodological foundation rather than a complete deployment-oriented validation. Due to time and budget constraints, we did not conduct a user study, but we consider a user-centered and practitioner-oriented evaluation an essential next step. Finally, our framework currently resolves disagreements at the *method-selection* level; it does not merge or reconcile feature-level disagreements for a single prediction, which remains an open avenue for future investigation.

# 8 Conclusion

This paper addressed the critical *disagreement problem* in XAI by proposing and validating a novel evaluation framework based on comparative Multi-Criteria Decision Analysis. By leveraging SAW, TOPSIS, and ELECTRE I, we moved beyond a single aggregated score to a transparent, preference-driven process for selecting the most suitable post-hoc explanation method. Validated on four ML/DL models, five explanation methods, and three real-world

datasets, our framework successfully navigates the trade-offs between competing explanation quality criteria (e.g., fidelity and stability), highlighting not only the best-performing methods under different scenarios but also the critical sensitivities in the decision logic itself. Overall, this work provides practitioners and researchers with a robust, adaptable methodology for structured, preference-aware explainer selection, supporting more principled, justifiable choices about explainability and fostering greater trust and reliability in AI systems.

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
