# OpenReview forum: "Resolving Disagreement Problems in Explainable Artifi- cial Intelligence Through Multi-Criteria Decision Analysis"
_TMLR — Rejected by TMLR_

### Review · Reviewer_Tpfw · 2026-02-16

**Summary Of Contributions:**

This paper proposes a simple strategy to choose between different post-hoc explanation methods when they disagree. The idea is to score each explainer (here: SHAP, LIME, Anchor) on a small set of explanation-quality metrics, let users assign weights associated with these metrics that reflect what they care about, then aggregate the scores using standard multi-criteria decision-making rules (SAW, TOPSIS, and ELECTRE I). Empirically, it reports explainer rankings under different weighting scenarios across a few tabular datasets and models, and highlights that both the chosen weights and the aggregation rule can change which explainer comes out on top.

The main weaknesses are that the paper feels incremental and a bit dated for a 2026 ML audience: it focuses on older XAI methods (LIME/SHAP/Anchor), classic tabular benchmarks, and classic backbone models, without a strong "why now" motivation, or evidence of downstream impact beyond producing rankings. Some components are also unclear or possibly incorrect (e.g., key metric definitions look internally inconsistent, and "ground-truth feature importance" for fidelity is not well-specified). The experiments are also fairly light (small sample sizes; limited model diversity), so the conclusions about robustness and practical usefulness read stronger than what the evidence really supports.

**Additional Comments:**

Other minor comments:
- the text is sometimes too generous with "state-of-the-art" mentions about quite old works/references/methods, and how SHAP/LIME/Anchor provide "detailed" explanations, how the tested models are "widely used ones", etc.;
- pretty much all comparative MCDA works are fairly old, would there be anything worth citing that also touches on modern decision problems with e.g. language model explanations?
- the citation/reference format needs to be cleaned up across the text.

**Audience:**

No

**Audience Explanation:**

I feel this paper would probably only interest a small slice of TMLR's audience, and mostly for fairly specific reasons.

On one hand, XAI is not going away; if anything it becomes more important as reasoning models are deployed in settings where people will demand auditability, contestability, and some notion of "why did it do that?". In that sense, a paper that tries to make explainer choice more systematic, and that highlights how rankings can flip depending on what you value, could be of some interest to practitioners and reviewers who care about evaluation methodology rather than new explainers.

On the other hand, I suspect most of TMLR's 2026 readership will find the findings here fairly limited or dated, because they largely repackage well-known components, and the empirical takeaways are not clearly tied to the problems the community is currently struggling with. The more compelling version of this story, for today’s audience, would be to transplant the same "metrics + decision framework" idea into the world of reasoning models: define and validate analogs of identity/separability/consistency/fidelity for reasoning traces and tool-augmented behavior (especially faithfulness), then compare modern strategies for explanations and monitoring (prompted self-explanations, post-hoc critics, attribution-over-rationales, logit-lens style probes, verifier models, etc.). That is both hard and timely; it is exactly why relatively few papers do it well. In contrast, taking a largely 2015-era XAI setup and showing a preference-weighted ranking over SHAP/LIME/Anchor is less likely to feel like something the median TMLR reader needs to know in 2026.

**Claims And Evidence:**

No

**Claims Explanation:**

The experiments support a narrower claim than the title/introduction suggest: if you score explainers on a set of metrics and aggregate with SAW/TOPSIS/ELECTRE, you can get different "best explainer" rankings depending on priorities and the aggregation rule. The submission often frames this as "resolving explanation disagreement", and the evidence does not convincingly support that stronger interpretation. In practice, the disagreement is mostly being pushed into an extra design choice: the user has to pick metric weights (and even an aggregation philosophy). Without a clear, validated way for users to set those weights, it is hard to argue the method resolves disagreement rather than re-labels it as preference selection.

Relatedly, the paper does not really show that users can set weights in a reliable, model- and dataset-aware way; nor does it evaluate whether different reasonable weight choices lead to stable selections, or whether the chosen explainer improves downstream outcomes (debugging, trust calibration, error detection, decision quality, etc.). So the evidence is clear enough for the mechanics of the framework, but not convincing for the central “resolution” claim and its practical implications.

**Requested Changes:**

1) Fix and fully specify the core metrics (critical): check the formal definitions vs the actual implementation for identity and separability, as the conditions read inverted for those two. Also, make sure there are no inconsistencies in norms anywhere, clearly define all distance functions, normalization, and hyperparameters (k, epsilon/neighborhood size, perturbation scheme, random seeds). For "fidelity", clarify what "ground-truth feature importance" means in practice; if it is model-derived, rename it and justify it.

2) Strengthen the empirical study to match the claims (critical): expand beyond a few tabular datasets and a small model set (known limitation); include more diverse datasets and modern model families. Increase sample sizes and include ablations/sensitivity analyses (how stable are scores and rankings as you vary k, epsilon, sampling, seeds, weights, etc.). Validate that the framework helps in practice (not just "rankings exist"); maybe consider a user study (or at least a practitioner-facing evaluation) showing the method improves something measurable: decision quality, debugging time, error detection, trust calibration, or choice consistency across users...

3) Clarify and narrow the central claim: reframe from "resolving disagreement" to "supporting structured explainer selection under preferences", unless you actually add a method that reconciles instance-level disagreements.

---

> ### Author Response · Authors · 2026-03-12
>
> **Concern 1:  The main weaknesses are that the paper feels incremental and a bit dated for a 2026 ML audience: ...... so the conclusions about robustness and practical usefulness read stronger than what the evidence really supports.**
>
> **Response:** We thank the reviewer for this comment. We respectfully disagree that the paper is dated merely because it evaluates LIME, SHAP, and Anchor on tabular benchmarks with standard predictive AI models. This experimental design followed the same setup commonly used in state-of-the-art XAI disagreement studies (Laberge et al., 2024; [1], [3], [4]), in which similar explanation methods, datasets, and backbone models are employed to address the XAI disagreement problem. Furthermore, recent state-of-the-art work at major AI conference venues such as ICLR, FAccT, and ICML [5–9], together with prior TMLR publications such as [3] and [4], showed that XAI, XAI evaluation, and explanation disagreement remain active and important research areas.
>
> It is worth mentioning that our goal in this work is not to introduce a new explanation method or benchmark, but to address a different, still unresolved problem: how to select the most suitable explanation method when multiple explanation methods disagree, and no single method performs best across all explanation evaluation metrics. This is precisely the motivation stated in our introduction: one explanation method may achieve high fidelity but poor stability, while another may be more robust at the expense of faithfulness, creating a genuine multi-criteria dilemma for practitioners. The key research gap, therefore, is not the absence of evaluation metrics but the absence of a principled decision-making framework that can synthesize these competing evaluation metrics in a transparent and context-sensitive manner. Our proposed MCDA framework is intended to fill that gap. We also respectfully showed that the contribution goes beyond merely producing rankings. The proposed framework makes explanation selection transparent under explicit user preferences and shows how different decision logics can surface different trade-offs among competing explanation properties. In this sense, the practical value of the work lies not only in ranking methods but also in supporting a justified, preference-aware explanation selection process when disagreement exists among candidate explanation methods. We explicitly stated that in the Introduction Section of the manuscript.
>
>
> We agree that the current manuscript can improve the clarity and precision of the evaluation-metric definitions. Importantly, our goal in this work was not to introduce new explanation-evaluation metrics, but to evaluate explanation disagreement using established metrics from prior XAI literature Barr et al. (2023); Agarwal et al. (2022); Dai et al. (2022); Bobek et al. (2021); Parimbelli et al. (2023); Solís-Martín et al. (2023); Klein et al. (2024). Therefore, the fidelity metric used in our study was adopted from prior work, similar to that of Dai et al. (2022), rather than being newly proposed here. We agree, however, that the current wording around “ground-truth feature importance” is not sufficiently precise and may be misleading. In the revised version, we will correct this description and revise these metrics more broadly so that the definitions and intuitions are internally consistent and clearly presented.
>
> We also appreciate the concern regarding the empirical scope. In the current version, we report results based on 50 randomly selected test samples for each dataset, model, and explainer combination in order to maintain a consistent and computationally manageable evaluation setting. At the same time, the study considers a diverse set of predictive models from both ML and DL, including RF, XGBoost, Logistic Regression, and a neural network, across three real-world datasets, consistent with prior work in the XAI disagreement literature Laberge et al. (2024); Han et al. (2022); Krishna et al. (2022), [1]. Nevertheless, we agree that the current experimental scale remains limited in terms of sample size and that the conclusions should be calibrated accordingly. Thus, in the revised version, we will extend the sample sizes and report the corresponding results. We believe this will provide a stronger empirical basis for the paper and allow a more convincing assessment of the framework’s robustness. We will also moderate the wording of our claims regarding robustness and practical usefulness to better align with the current evidence, positioning the present study as a controlled validation of the proposed framework rather than a definitive empirical conclusion.

---

> > ### Author Response · Authors · 2026-03-12
> >
> > **Concern 4: I feel this paper would probably only interest a small slice of TMLR's audience...**
> >
> > **Response:** We agree that the paper is likely to be of strongest interest to the segment of the TMLR community working on XAI, XAI evaluation, and explanation disagreement. However, we respectfully believe that this still represents a meaningful and appropriate audience for TMLR, especially since our contribution is methodological rather than narrowly tied to a single application domain. Moreover, prior work on XAI and explanation disagreement has also appeared in TMLR, including works such as [3] and [4], which further indicates that these topics are well within the journal’s scope and of sustained interest to its readership.This paper addresses a practical and increasingly important question: when multiple post-hoc explainers provide conflicting explanations for the same prediction, how should a practitioner decide which explainer to trust? As we motivate in the paper, such disagreement can directly affect auditability, contestability, trust, and downstream decision-making in high-stakes domains. In this sense, the core gap is not the absence of evaluation metrics, but the absence of a structured process for synthesizing them when no single explainer is uniformly best. Our framework is intended to fill precisely this gap by formalizing explainer choice as a transparent and preference-aware decision problem.
> >
> > We also believe that the paper is relevant beyond a narrow niche because it speaks to both practitioners and researchers. For practitioners and domain experts, the framework provides a systematic way to choose among competing explanation methods based on explicit priorities, such as fidelity, stability, or faithfulness, rather than relying on default choices, anecdotal preferences, or method popularity. For researchers, the paper contributes to XAI evaluation methodology by showing that explainer rankings depend not only on the underlying explanation evaluation metrics but also on the aggregation logic used to combine them. In particular, the differences we observe between compensatory methods such as SAW and TOPSIS and the non-compensatory ELECTRE I method suggest that aggregation itself is an important and often overlooked design choice in XAI meta-evaluation. This is exactly the type of methodological issue that we believe is relevant to a broader ML audience interested in how evaluation frameworks should be designed and interpreted. We already discussed this in the Discussion Section of our manuscript.
> >
> > More broadly, we do not view this as a problem of interest only to a very small audience, because explanation disagreement has already emerged as a recognized research topic in the recent XAI literature. Prior work, including Krishna et al. (2022), Han et al. (2022), Silva et al. (2025), and Laberge et al. (2024) [1], shows that disagreement between explainers is not a minor artifact, but a substantive challenge for the reliable use and evaluation of XAI methods. In particular, [3] and [4], which were also published in TMLR, likewise address problems related to XAI disagreement and the broader challenge of making explanations more reliable and useful. Our contribution complements this line of work by focusing on the decision problem that follows from such disagreement: how to select the most suitable explainer under explicit user preferences and competing quality criteria.
> >
> > We respectfully disagree with the suggestion that XAI is losing relevance. On the contrary, recent state-of-the-art work at major AI conference venues such as ICLR, FAccT, and ICML [5–9], together with prior TMLR publications such as [3] and [4], showed that XAI, XAI evaluation, and explanation disagreement remain active and important research areas. In particular, [3] and [4] further demonstrate that improving the reliability of explanations and addressing disagreement across explanation methods are already recognized as meaningful problems within the TMLR community. In this context, our paper contributes to an ongoing and growing line of research by focusing on the decision problem that naturally arises once explanation disagreement is acknowledged as a persistent feature of XAI systems.
> >
> > As reasoning-oriented models are deployed in settings requiring auditability and clear justification, the challenge is not only generating explanations but choosing among conflicting explanations from different post-hoc methods. Our paper addresses this by formalizing explainer selection as a transparent, preference-aware decision problem, rather than assuming a universally best method. While validated on established ML/DL models and benchmark datasets, we did not evaluate the framework on large-scale foundation models due to time and resource constraints. This limitation and the extension to Transformer-based LLMs and diffusion models are already noted in the paper’s **Limitations and Future Work** section.

---

> ### Author Response · Authors · 2026-03-12
>
> **References**
>
> *[1] Multi-criteria Rank-based Aggregation for Explainable AI, https://arxiv.org/abs/2505.24612*
>
> *[3] Laberge, G., Batiste Pequignot, Y., Marchand, M. \&amp; Khomh, F.. (2024). Tackling the XAI Disagreement Problem with Regional Explanations. <i>Proceedings of The 27th International Conference on Artificial Intelligence and Statistics</i>, in <i>Proceedings of Machine Learning Research</i> 238:2017-2025 Available from https://proceedings.mlr.press/v238/laberge24a.html.*
>
> *[4] Dubé, G., & Marchand, M. (2025). Shapley Values of Structured Additive Regression Models and Application to RKHS Weightings of Functions. Transactions on Machine Learning Research.*
>
> *[5] Laberge, G., \& Ahmad, O. Tackling the XAI Disagreement Problem with Adaptive Feature Grouping. In The Fourteenth International Conference on Learning Representations.*
>
> *[6] Hussain, A., Thacker, C. M., Zhao, P., Karan, A., \& Vincent, N. Empowering Users Together: Connecting Algorithmic Collective Action and Explainable AI. In NeurIPS 2025 Workshop on Algorithmic Collective Action.*
>
> *[7] Rawal, K., Fu, Z., Delaney, E., \& Russell, C. (2025, June). Evaluating model explanations without ground truth. In Proceedings of the 2025 ACM Conference on Fairness, Accountability, and Transparency (pp. 3400-3411)*
>
> *[8] Lee, J. R., Emami, S., Hollins, M. D., Wong, T. C., Villalobos Sánchez, C. I., Toni, F., ... \& Dejl, A. (2025, June). Xai-units: Benchmarking explainability methods with unit tests. In Proceedings of the 2025 ACM Conference on Fairness, Accountability, and Transparency (pp. 2892-2905).*
>
> *[9] Bordt, S., Raidl, E., \& Von Luxburg, U. (2025, October). Position: Rethinking Explainable Machine Learning as Applied Statistics. In Forty-second International Conference on Machine Learning Position Paper Track"*

---

> ### Author Response · Authors · 2026-03-12
>
> **Concern 2: The experiments support a narrower claim than the title/introduction suggest: if you score explainers on a set of metrics and aggregate with SAW/TOPSIS/ELECTRE, you can get different "best explainer" rankings depending on priorities and the aggregation rule. The submission often frames this as "resolving explanation disagreement", and the evidence does not convincingly support that stronger interpretation. In practice, the disagreement is mostly being pushed into an extra design choice: the user has to pick metric weights (and even an aggregation philosophy). Without a clear, validated way for users to set those weights, it is hard to argue the method resolves disagreement rather than re-labels it as preference selection.**
>
> **Response:** We thank the reviewer for this comment, which gets to the heart of our paper's contribution. The reviewer correctly points out that our framework does not eliminate disagreement in an absolute sense, but rather pushes the choice to a higher level of abstraction: selecting metric weights and an aggregation philosophy. We agree with this characterization and believe this transformation is precisely where the novelty and utility of our work lie. Our goal is not to find a single, objective "winner" where none exists, but to provide a principled methodology for navigating these disagreements. We argue that ``resolving" the disagreement problem in practice is not about finding a single ground truth, but about making a *defensible and transparent choice* in the face of conflicting evidence. Our framework contributes to this resolution in three key ways:
>
> **From Ambiguity to Structure:** Without our framework, a practitioner faced with conflicting explanations from for example: LIME and SHAP and is left with an unstructured, qualitative dilemma. Our method transforms this ambiguous disagreement into a structured **MCDA** problem. By externalizing the decision criteria (our six metrics) and forcing a user to articulate their priorities (the weights), we replace an intractable, implicit trade-off with a formal, explicit one. This structuring is, in itself, a form of resolution.
>
> **Making Trade-offs Explicit and Quantifiable:** The core issue with raw explainer disagreement is that the trade-offs are hidden. A user doesn't know \textbf{why} SHAP and LIME disagree or what they would be giving up by choosing one over the other. Our framework makes these trade-offs explicit. For example, our results show that choosing LIME under a 'Stability-Focused' scenario (where it often wins on compensatory methods like SAW/TOPSIS) comes at a quantifiable cost to the `Identity' metric. The non-compensatory ELECTRE~I method makes this trade-off even more salient by flagging LIME and SHAP as incomparable. This insight---that a choice is not "right" or ``wrong" but a quantifiable trade-off---is the resolution we provide.
>
> **Enabling Justifiable and Auditable Decisions:** By re-labeling disagreement as "preference selection", our framework provides a path to a justifiable decision. In a high-stakes domain like finance or healthcare, a practitioner can now document their choice: *``We selected LIME for this task because our primary requirement, as documented by our `Stability-Focused' weight vector, is robustness to input perturbations. Our analysis showed that while SHAP is a strong generalist, LIME is the superior choice for this specific, stated priority."* This transforms a subjective guess into an auditable, principled decision. We have clarified this in the paper's introduction and discussion to better frame our contribution not as finding a single truth, but as providing a framework for defensible decision-making.
>
> We do not claim that our work make the disagreement vanish. Instead, it re-frames it as a preference-elicitation and decision-making problem, providing the first structured, multi-logic framework to resolve it in a transparent, quantifiable, and ultimately more practical way. We have revised the text to make this framing clearer, emphasizing our contribution as a "principled decision-making framework for navigating XAI disagreement" rather than a method that "eliminates" it.

---

> ### Author Response · Authors · 2026-03-12
>
> **Concern 3: Relatedly, the paper does not really show that users can set weights in a reliable, model- and dataset-aware way; nor does it evaluate whether different reasonable weight choices lead to stable selections, or whether the chosen explainer improves downstream outcomes (debugging, trust calibration, error detection, decision quality, etc.). So the evidence is clear enough for the mechanics of the framework, but not convincing for the central “resolution” claim and its practical implications.**
>
> **Response:**  We thank the reviewer for this comment. Our primary goal in this paper is to propose and technically validate the mechanics of the comparative MCDA framework for addressing explanation disagreement. More specifically, we show that the framework provides a feasible, flexible, and transparent way to compare explanation methods across multiple evaluation criteria, and that it reveals how explainer rankings change as user priorities and aggregation logics vary. Our intention in this first step was to establish the soundness of the decision-analytic approach before moving to user-centered validation and downstream task-based evaluation. In this sense, the current paper should be viewed as a methodological foundation rather than a complete end-to-end validation of practical deployment. While we agree that the current manuscript does not yet demonstrate that users can set weights in a fully reliable, model- and dataset-aware manner, nor does it evaluate whether the selected explainer improves downstream outcomes such as debugging, trust calibration, error detection, or decision quality. Due to time and budget constraints, our current work did not include a user study to address these questions. We consider this an important direction for future research and will explicitly include this point in the Limitations and Future Work section of the revised manuscript.
>
> Regarding weight elicitation, the predefined weighting scenarios in the current manuscript were intended as illustrative proxies for different user priorities, rather than as a claim that real users can already specify such weights in a fully reliable or context-aware way. We agree that a more systematic and user-friendly approach is needed for practical application. A promising next step is to integrate structured preference-elicitation techniques, such as the Analytic Hierarchy Process (AHP), which can derive more consistent weight vectors through pairwise comparisons of criteria. This would make the process of setting weights less arbitrary and more interpretable for practitioners. Due to time and budget constraints, however, the current submission did not include a user study or formal elicitation procedure to address this issue.
>
> Similarly, we agree that the practical implications of the framework should ultimately be validated through downstream task-based evaluation. Assessing whether the selected explainer improves outcomes such as trust calibration, debugging efficiency, error detection, or decision quality would require a carefully designed human-subject or practitioner-facing study with clearly defined tasks and measurable performance criteria. We consider this an important next step, but it is also a substantial research effort in its own right. Our current paper provides the foundational selection mechanism; follow-up work will be needed to evaluate its downstream utility with human participants.
>
> We will revise the manuscript to make this scope clearer. In particular, we will explicitly state in the Limitations and Future Work section that reliable user-centered weight elicitation and downstream validation are not addressed in the current submission and remain key directions for future work. We will also clarify that the present evidence supports the mechanics of the framework, while the stronger claim of practical “resolution” requires further validation through user studies and task-based evaluation.

---

> ### Author Response · Authors · 2026-03-12
>
> **Concern 5: On the other hand, I suspect most of TMLR's 2026 readership will find the findings here fairly limited or dated, because they largely repackage well-known components, and the empirical takeaways are not clearly tied to the problems the community is currently struggling with. The more compelling version of this story, for today’s audience, would be to transplant the same "metrics + decision framework" idea into the world of reasoning models:...**
>
> **Response:** We thank the reviewer for this comment. Our current submission is a foundational methodological contribution rather than a final answer to the broader agenda of explainability for reasoning models. The paper addresses a persistent, still practically important problem for a large class of currently deployed AI systems: when multiple post-hoc explanation methods disagree, how should a practitioner systematically choose among them given explicit user preferences? As we state in the introduction, the central gap is not the absence of evaluation metrics, but the absence of a structured decision-making process for synthesizing them when no single explainer is uniformly best. Our contribution is to formalize this problem through an MCDA framework and to show that different decision logics can reveal materially different trade-offs. We also do not view the focus on LIME, SHAP, and Anchor as merely dated; rather, it is a deliberate choice of a well-understood and practically relevant testbed. These methods remain among the most widely used and studied post-hoc explainers, and the disagreement problem among them is still unresolved for practitioners working with tabular and conventional predictive models in domains such as healthcare, finance, and risk assessment. The manuscript explicitly motivates this through high-stakes deployment scenarios in which conflicting explanations can undermine trust, auditability, and downstream decision-making. More broadly, the paper is situated within an active research literature on explanation disagreement, and recent work at major venues such as ICLR, FAccT, and ICML [5–9], together with prior TMLR publications such as [3] and [4], further shows that XAI, XAI evaluation, and explanation disagreement remain active and important research areas.
>
> More importantly, we believe the paper's main contribution is not the specific ranking of SHAP, LIME, and Anchor, but rather the methodological blueprint for preference-aware explainer evaluation. The framework is intentionally comparative: it combines six explanation-quality criteria, organizes them across model alignment, robustness, and inter-method agreement, and contrasts compensatory and non-compensatory MCDA logics. The empirical analysis shows that this distinction matters in practice. For example, the manuscript reports cases in which SAW and TOPSIS rank LIME first, whereas ELECTRE I keeps LIME and SHAP jointly in the kernel because gains on a prioritized criterion are offset by weaknesses on other criteria. In this sense, the paper’s core contribution is to expose trade-offs that simpler ranking schemes would hide, and we view this decision-theoretic structure as domain-agnostic and therefore relevant beyond the specific explainer family studied here.
>
>
> At the same time, we also agree with the reviewer that a highly compelling extension of the present work would be to transplant the same “metrics + decision framework” perspective to modern reasoning models (e.g., large language models (LLMs), tool-augmented systems, and explanation or monitoring strategies designed for those settings. In particular, defining and validating analogues of identity, separability, consistency, fidelity, and especially faithfulness for reasoning traces would be both timely and challenging, and we agree that this is an important direction for the field. Due to time and resource constraints, we did not evaluate the framework on more recent large-scale foundation-model settings in the current submission. However, we already state this explicitly in the Limitations and Future Work section, where we said that extending the framework to more complex generative AI architectures, including Transformer-based models that underpin large language models and reasoning systems, as well as diffusion models, is an important direction for future work. We will make this point even clearer in the revision to ensure the paper's current scope and intended next steps are communicated more precisely.

---

> ### Author Response · Authors · 2026-03-12
>
> **Concern 6: Fix and fully specify the core metrics (critical): check the formal definitions vs the actual implementation for identity and separability, as the conditions read inverted for those two. Also, make sure there are no inconsistencies in norms anywhere, clearly define all distance functions, normalization, and hyperparameters (k, epsilon/neighborhood size, perturbation scheme, random seeds). For "fidelity", clarify what "ground-truth feature importance" means in practice; if it is model-derived, rename it and justify it.**
>
> **Response:** We thank the reviewer for the valuable feedback. We will address the reviewer's concerns regarding methodological clarity and reproducibility by making the following corrections and additions to the manuscript.
>
> ### a) Corrected Metric Definitions
> Regarding the formal definitions for 'Identity' and 'Separability', we thank the reviewer for pointing this out. In this work, our goal was not to introduce a new explanation evaluation metric, such as a new Identity metric, but rather to evaluate explanation disagreement using established state-of-the-art evaluation metrics from prior work. Therefore, we adopted the same Identity and Separability metric formulation used in the existing literature, similar to Solís-Martín et al. (2023).
>
> ### b) Clarified "Ground Truth" for Fidelity
> The term "ground-truth feature importance" is ambiguous. We will revise the text to state that we use the global SHAP feature importance over the test set as a consistent and deterministic *proxy* for this reference vector. We will acknowledge the potential for a slight bias towards SHAP on this metric in our limitations section.
>
> ### c) Specified Hyperparameters and Implementation Details
> Key details were missing. We will added a new "Implementation Details" appendix, referenced from the main text, that will specifies all parameters:
>
> - **General:** We use **k = 5** for all top-k sets. The distance function 'd(·,·)' is the Euclidean ('L₂') distance.
> - **Stability:** The neighborhood radius is **ε = 0.1**, with neighbors generated via Gaussian noise 'N(0, 0.05²)`.
> - **Faithfulness (PGI):** We use **p = 10** perturbation runs per instance.
> - **Consistency Aggregation:** The final score for a method is its average consistency against all others (e.g., 'CO(SHAP) = avg(CO(SHAP,LIME), CO(SHAP,Anchor))`).
> - **Anchor to Vector Mapping:** We now specify that Anchor rules are converted to a feature importance vector by assigning the rule's precision as the importance for features within the rule and zero for all others.
>
> ### d) Formalized Normalization
> The normalization process was not explicitly defined. We will added a "Formal Preliminaries" subsection that provides the exact formulas used. We first transform cost metrics (Stability, Separability) to benefit metrics via the standard formula:
>
> $x''_{ij} = (\max_k(x'_{ik}) - x'_{ij}) / (\max_k(x'_{ik}) - \min_k(x'_{ik}))$.
>
> Subsequently, all metrics are scaled to a '[0,1]` range using min-max normalization.
>
> These revisions ensure our methodology is now fully specified, correct, and reproducible.

---

> ### Author Response · Authors · 2026-03-12
>
> **Concern 7: Strengthen the empirical study to match the claims (critical): expand beyond a few tabular datasets and a small model set (known limitation); include more diverse datasets and modern model families. Increase sample sizes and include ablations/sensitivity analyses (how stable are scores and rankings as you vary k, epsilon, sampling, seeds, weights, etc.). Validate that the framework helps in practice (not just "rankings exist"); maybe consider a user study (or at least a practitioner-facing evaluation) showing the method improves something measurable: decision quality, debugging time, error detection, trust calibration, or choice consistency across users.**
>
> **Response:** In the current manuscript, the evaluation is conducted on three real-world tabular datasets, four predictive models, and three post-hoc explanation methods, with explanation metrics computed on 50 randomly selected test instances per dataset. state-of-the-art post-hoc explanation methods using six state-of-the-art evaluation metrics across three widely used open-source benchmark datasets, and we analyzed their behavior across multiple predictive AI models, following evaluation protocols similar to those adopted in prior work on XAI disagreement, such as Han et al. (2022) Krishna et al. (2022), [1].  However, we agree that expanding the evaluation set size would further strengthen the empirical validation of our framework. Therefore, in the revised version, we will extend the test set size and report the corresponding results. We believe that this addition will provide a stronger empirical basis for our conclusions and further demonstrate the robustness of the proposed framework. Due to time and budgetary constraints, we did not evaluate the framework on more recent large-scale foundation-model settings in the current submission. However, we already explicitly explained this in the Limitations and Future Work section, stating that extending the framework to more complex generative AI architectures, including Transformer-based models that underpin large language models and reasoning systems, as well as diffusion models, is an important direction for future work. We also agree with the reviewer that practical usefulness should ultimately be validated more directly than by showing that rankings can be produced. In this submission, we aim to provide a principled, transparent framework for selecting explainers based on multiple criteria and user preferences. However, due to time and budgetary constraints, we were unable to do full user-centered validation via a user study. Therefore, in the future, we plan to assess the proposed framework through a user study or practitioner-oriented evaluation that examines measurable outcomes such as decision quality, debugging time, error detection, trust calibration, or the consistency of explainer choice across users. We will make this future direction more explicit in the Limitations and Future Work section in our revised manuscript.
>
> **Concern 8: Clarify and narrow the central claim: reframe from "resolving disagreement" to "supporting structured explainer selection under preferences", unless you actually add a method that reconciles instance-level disagreements.**
>
> **Response:** We thank the reviewer for this comment. We will revise the manuscript's title, abstract, introduction, and conclusion to reframe our central claim from "resolving disagreement" to "providing a principled framework for structured explainer selection based on user preferences" in our revised version. This more accurately reflects our contribution: transforming unstructured disagreement into a transparent, justifiable decision-making process rather than reconciling instance-level attribution differences.

---

> ### Author Response · Authors · 2026-03-12
>
> **Concern 9: Other minor comments:**
>
> - The text is sometimes too generous with "state-of-the-art" mentions about quite old works/references/methods, and how SHAP/LIME/Anchor provide "detailed" explanations, how the tested models are "widely used ones", etc.;
> - Pretty much all comparative MCDA works are fairly old, would there be anything worth citing that also touches on modern decision problems with e.g. language model explanations?
> - The citation/reference format needs to be standardized throughout the text.
>
> **Response:** We thank the reviewer for this comment.
>
> - We will perform a thorough review of the manuscript and will remove or tone down subjective descriptors like "state-of-the-art" etc where they are not strictly necessary or could be seen as overstatement. The focus is now on more objective descriptions of the methods and their properties.
>
> - We thank the reviewer for this forward-looking suggestion. While the foundational MCDA methods (SAW, TOPSIS, ELECTRE) are indeed well-established, their application to modern AI explainability, particularly for navigating trade-offs between quantitative XAI metrics, is novel. We have searched for and were unable to find existing work applying a comparative MCDA framework to the selection of explainers for language models. However, we will explicitly positioned this as a critical direction for future work in our revised **Limitations and Future Work** section, acknowledging the timeliness of the reviewer's point.
>
> - We will perform a complete pass over the manuscript and the bibliography to ensure that all citations and references are standardized and adhere to the required format. All inconsistencies will be resolved.

---

### Review · Reviewer_8psN · 2026-02-24

**Summary Of Contributions:**

This paper studies the explanation disagreement problem in post-hoc explainable AI (XAI), where different explanation methods (e.g., SHAP, LIME, Anchor) produce conflicting feature attributions for the same prediction. The authors argue that the core issue is not identifying a single best explainer, but selecting an explainer appropriate to user priorities. They propose to formulate explanation selection as a Multi-Criteria Decision Analysis (MCDA) problem. The main claim: instead of choosing a universal best explainer, the framework selects the most suitable explainer for a given context, and reveals trade-offs.

Main contributions are as follws

1. Correctly identifies an important XAI problem: explanation disagreement undermines trust
2. Confirms no single explainer dominates all metrics

**Audience:**

Yes

**Audience Explanation:**

This paper applied a structured decision framework to resolve the disagreement issue across different explainers. This may be of interest to researchers in the XAI community.

**Claims And Evidence:**

No

**Claims Explanation:**

1. The work does not demonstrate any downstream impact of the proposed approach, such as improved alignment with human judgments or improved predictive performance.

2. No Comparison to explanation aggregation/ensemble methods [1],[2]



[1] Multi-criteria Rank-based Aggregation for Explainable AI, https://arxiv.org/abs/2505.24612
[2] O. Mitrut ,, G. Moise, A. Moldoveanu, F. Moldoveanu, M. Leordeanu, and L. Petrescu, “Clarity in complexity: how aggregating explanations resolves the disagreement problem,” Artificial Intelligence Review,

**Requested Changes:**

1. Include the ensemble approaches discussed above as baselines for comparison.
2. Provide appropriate statistical significance testing for the reported results.
3. Expanding the evaluation set size would further strengthen the empirical validation.

---

> ### Author Response · Authors · 2026-03-12
>
> **Concern 1: 1. The work does not demonstrate any downstream impact of the proposed approach, such as improved alignment with human judgments or improved predictive performance.**
>
> **2. No Comparison to explanation aggregation/ensemble methods [1],[2].**
>
> *[1] Multi-criteria Rank-based Aggregation for Explainable AI, https://arxiv.org/abs/2505.24612*
>
> *[2] O. Mitrut ,, G. Moise, A. Moldoveanu, F. Moldoveanu, M. Leordeanu, and L. Petrescu, “Clarity in complexity: how aggregating explanations resolves the disagreement problem,” Artificial Intelligence Review*
>
> **Response** We thank the reviewer for this comment. Our paper’s primary objective is to address the disagreement problem in explainable AI (XAI) at the explanation-method selection stage. This is indeed an important problem that has received much attention from the community in the last few years [1, 3, 5]. Due to the nature of the XAI disagreement problem, improved predictive AI model accuracy is neither a direct nor an expected outcome of the proposed framework, since the underlying AI model is not modified. Instead, our contribution is to provide a more transparent, preference-aware, and justifiable process for selecting an explanation method in settings where disagreement among explanation methods creates uncertainty for researchers and practitioners. In such cases, the practical value of our approach lies in helping users choose the explanation method that best aligns with their priorities across multiple evaluation criteria, rather than in improving the predictive model itself. Therefore, demonstrating downstream human-centered impact, such as alignment with human judgments, user confidence, or decision consistency, is beyond the scope of the present work.
>
> We would like to emphasize that the reviewer provided literature [1] and [2] actually addresses a different approach to solve the explanation disagreement problem than the one considered in our paper. In particular, [1] proposed a multi-criteria rank-based aggregation framework that combines several quality criteria to construct an ensemble explanation from multiple explanation methods. In contrast, our proposed MCDA framework is designed to select the most suitable explainer rather than to aggregate explainers. More specifically, our goal is to identify the most suitable explanation method from a set of candidate explainer under different user-priority scenarios, such as prioritizing fidelity, stability, or other evaluation criteria.  Although both works rely on multi-criteria reasoning, their objectives are fundamentally different: [1] aggregates explanation outputs, whereas our framework ranks and selects the most appropriate explanation method from a set of candidate methods. A similar distinction applies to [2]. This work investigates whether aggregating explanations can mitigate the Rashômon effect and reduce disagreement across attribution methods by producing a consensus explanation. However, its problem setting, methodology, and evaluation objective differ substantially from ours. In particular, [2] does not assess disagreement using the explanation-quality criteria that underpin our framework, such as fidelity, stability, identity, separability, consistency, and faithfulness. Instead, it focuses on the behavior of aggregated explanations and the downstream effects of combining explanation weights. For this reason, the results reported in [2] are not directly comparable to ours, as the paper addresses different decision problems and uses different evaluation protocols. Therefore, we were unable to treat [1] and [2] as direct baselines for empirical comparison, since our paper addresses explanation disagreement at the level of explainer selection and aims to determine which explanation method is most appropriate under explicit user preferences.

---

> ### Author Response · Authors · 2026-03-12
>
> **Concern 2: Include the ensemble approaches discussed above as baselines for comparison.**
>
> **Response:** For this, please see the responses above (Concern 1).
>
> **Concern 3: Provide appropriate statistical significance testing for the reported results.**
>
> **Response:** We thank the reviewer for this comment. In the current manuscript, Tables 1, 2, and 3 report the descriptive statistics, such as mean and standard deviation, over 50 randomly selected test samples for each dataset, model, and explainer combination, following common practice in prior work on XAI disagreement literature  Han et al. (2022); Krishna et al. (2022). We did not include formal statistical significance testing in the current version because our primary goal was to establish the existence of consistent multi-metric trade-offs across explainers and to use those aggregated metric profiles as inputs to the MCDA-based selection framework, rather than to make isolated pairwise inferential claims for each metric separately. This design choice aligns with the paper's overall objective: to support explainer selection across multiple criteria and user-defined preferences. Therefore, in the current submission, we limited the analysis to descriptive statistics and did not perform statistical significance testing on the per-instance metric values.
>
> **Concern 4: Expanding the evaluation set size would further strengthen the empirical validation.**
>
> **Response:** We thank the reviewer for this suggestion. We agree that expanding the evaluation set size would further strengthen the empirical validation of our framework. In the current version, our experiments used 50 randomly selected test samples per dataset, model, and explainer combination to ensure a consistent, computationally manageable evaluation setting. In the revised version, we will extend the test set size (e.g., compute results for the full test set) and report the corresponding results. We believe that this addition will provide a stronger empirical basis for our conclusions and further demonstrate the robustness of the proposed framework.

---

### Review · Reviewer_dRQ4 · 2026-02-26

**Summary Of Contributions:**

This paper deals with tackling the disagreement problem amongst post-hoc explainers and answers the question of which specific post-hoc explanation methods to use and when.

They pose the problem of selecting an explanation method given user preference as a Multi-Criteria Decision Analysis(MCDA) problem across six metrics for assessing the explanation quality. They use established techniques of MCDA namely Simple Additive Weighting, TOPSIS, ELECTRE I to aggregate these evaluations and rank them based on user-defined priorities.

Strengths: The paper tackles the important problem of disagreement among post-hoc explainers and proposes an MCDA-based framework to guide explainer selection under user preferences. The idea of moving beyond a single “best” explainer toward preference-aware aggregation is conceptually well-motivated and potentially valuable for practical deployment.

Weaknesses: However, the current version does not provide sufficiently rigorous empirical support. Some evaluation metrics appear incorrectly specified or insufficiently justified. The experimental study lacks depth, particularly in demonstrating disagreement across closely related explainer variants and in showcasing the benefits of MCDA under genuinely non-trivial user preference settings. Consequently, the practical utility of the proposed framework is not yet convincingly established.

**Additional Comments:**

None

**Audience:**

Yes

**Audience Explanation:**

The problem of disagreement amongst explainers is an important one. The method helps the audience by providing a simple mechanism for picking which explainer to use and when. However, there is scope for improvement to prove usability.

**Claims And Evidence:**

No

**Claims Explanation:**

The paper's claims are not adequately supported by clear and convincing evidence for the following reasons: (1) Two of the core evaluation metrics are either incorrectly specified or insufficiently justified. (2) Limited experimental validation is done on explainers’ variants. (3) There is a lack of empirical studies under non-trivial user preference settings. Please look at requested changes for details.

**Requested Changes:**

(a) Changes that are critical to securing my recommendation for acceptance:

1. Section 3.2, Page 6 - Identity: I feel that the given equation is incorrect and should be ∀i,j(d(x_i, x_j)=0 => d(ε_i, ε_j)=0). Please comment and correct, if needed. Also ensure that the correct equation has been used to compute the empirical results.

2. Section 3.2, Page 6 - Separability: An explanation about why separability should be low must be added. At first read, it may confuse the readers since the definition and the equation do not match.

3. Add experiments on different variants of the same class of explainers, such as LIME, BayesLIME, etc. Showcase how much disagreement exists among them and use your MCDA formulation for ranking amongst them. Give insights on why, for a particular dataset, one variant is better than the other.

4. The authors have motivated and given definitions of the various MCDA aggregation schemes with a weight vector. Yet, when we look at the rankings in Table 4, they seem to be addressing a single metric-focused aggregation, which any user can simply decipher manually upon looking at the metrics in the tables corresponding to various datasets, rendering the MCDA-based aggregation mechanism moot. The authors are requested to design experiments and showcase the utility of their method in the presence of non-trivial preferences.

(b) Minor corrections that would simply strengthen the work.

1. Page 7: Line numbers of all algorithms seem to be incorrectly marked as 0. Please correct.

2. In Alg 1 and Alg 3, return statements are suffixed by an =0 output. This seems to be a typo. Please correct.

3. The ordering of the algorithms is 2, 1, 3. Please ensure it is in the correct order: Alg 1, 2, 3.

---

> ### Author Response · Authors · 2026-03-12
>
> **Concern 1: Weaknesses: However, the current version does not provide sufficiently rigorous empirical support...... of the proposed framework is not yet convincingly established.**
>
> **Response:** We thank the reviewer for this comment. We will strengthen empirical supports in the revised version by increasing the sample size and including variants from the same explanation method class (e.g., LIME-type methods such as BayesLIME and SHAP-type methods such as BayesSHAP). We also would like to clarify that this work evaluates three widely used state-of-the-art post-hoc explanation methods using six state-of-the-art evaluation metrics across three widely used open-source benchmark datasets, and we analyzed their behavior across multiple predictive AI models, following evaluation protocols similar to those adopted in prior work on XAI disagreement, such as [1]. Therefore, our experimental design is well aligned with the existing literature and follows a comparable validation strategy for studying disagreement across explanation methods. This design allows us to study explanation disagreement not as an isolated case, but as a recurring phenomenon across datasets, model classes, and evaluation criteria.
> Regarding evaluation metrics, we would like to clarify that the goal of this work is not to introduce new explanation evaluation metrics, but rather to study explanation disagreement using established state-of-the-art explanation evaluation metrics from prior literature. For instance, regarding the Identity metric, we adopted the formulation used in existing work, consistent with prior studies such as Solís-Martín et al. (2023). However, during the initial submission, we mistakenly provided an incorrect citation for this metric. We will correct this in the revised manuscript so that the presentation accurately reflects the prior work on which our evaluation protocol is based. At the same time, we agree that the current presentation of some metrics can be made clearer and better justified. For instance, the description of the Separability metric in the submitted version may be confusing because the textual explanation and the formal expression are not clearly aligned. We will revise the explanation evaluation metric section of the manuscript to make the intended interpretation explicit and ensure that the definition, equation, and accompanying discussion are fully consistent.
>
> We agree that evaluating closely related variants within the same explainer family would strengthen the empirical study. Therefore, in the revised version, we will extend the experiments to include variants from the same explainer class, specifically LIME-type methods such as BayesLIME and SHAP-type methods such as BayesSHAP, and we will evaluate them using the same six explanation evaluation metrics and the same MCDA pipeline. This will allow us to quantify the extent of disagreement even among conceptually similar explainer and to rank them under a common evaluation setting. It will also enable us to provide more concrete insights into why one variant may be preferred over another for a particular dataset, for example, whether a Bayesian variant improves robustness or identity while introducing trade-offs in fidelity or faithfulness. We believe this extension will strengthen the practical relevance of our proposed framework by showing that MCDA is useful not only for comparing substantially different explainer but also for distinguishing between closely related explainer variants where manual comparison is less straightforward.
>
> We want to clarify that the intended practical contribution of our work is not to claim a universally best explanation method but to provide a transparent and systematic decision-support framework for selecting the most suitable explanation method under explicit user preferences. In many realistic settings (e.g., healthcare, finance, law), practitioners face multiple competing explanation methods that differ along criteria such as fidelity, stability, identity, separability, consistency, and faithfulness, with no single method dominating across all of them. Our framework is designed precisely for this setting: it makes these trade-offs explicit and enables users to incorporate their priorities into the selection process, rather than relying on ad hoc choices or the default popularity of a particular method. In this sense, the practical utility of the framework lies in supporting principled explanation-method selection in the face of disagreement, rather than asserting that one explanation method should always be preferred. We will revise the manuscript to make this motivation and practical value clearer, while remaining faithful to the main goal of this work: resolving disagreement among explanation methods through a transparent, preference-aware MCDA formulation rather than ad hoc or purely manual comparisons.
>
> *[1] Multi-criteria Rank-based Aggregation for Explainable AI, https://arxiv.org/abs/2505.24612*

---

> ### Author Response · Authors · 2026-03-12
>
> **Concern 2: Requested Changes: (a) Changes that are critical to securing my recommendation for acceptance...**
>
> **Response:** Regarding the Identity Metric, we thank the reviewer for pointing this out. In this work, our goal was not to introduce a new explanation evaluation metric, such as a new Identity metric, but rather to evaluate explanation disagreement using established state-of-the-art evaluation metrics from prior work. Therefore, we adopted the same Identity metric formulation used in the existing literature, similar to Solís-Martín et al. (2023). However, during the initial submission, we mistakenly included an incorrect citation for this metric. We will correct this in the revised version so that the manuscript accurately reflects the prior work on which our proposed evaluation protocol is based.
>
> Regarding the Separability Metric, we agree that the current presentation of the Separability metric in the submitted manuscript may be unclear, because the textual explanation and the formal expression are not aligned as clearly as they should be. The main idea of the Separability metric in our paper is that non-identical samples should receive different explanations. Therefore, lower Separability values are preferred, as they indicate fewer cases in which distinct samples are assigned the same explanation. More specifically, if a feature is not actually relevant to the model’s prediction, then two samples that differ only in that feature may still lead to the same prediction. In such a situation, the explanation method may also return identical explanations even when the underlying inputs differ. We will revise this part of the manuscript to make this interpretation more explicit and to ensure that the definition, equation, and accompanying discussion are fully consistent in the revised version.
>
> We agree that evaluating closely related variants within the same explainer family would strengthen the empirical study. Therefore, in the revised version, we will extend the experiments and evaluate them using the same six explanation evaluation metrics and the same MCDA pipeline. This will allow us to quantify the extent of disagreement even among conceptually similar explainer and to rank them under a common evaluation setting. It will also enable us to provide more concrete insights into why one variant may be preferred over another for a particular dataset, for example, whether a Bayesian variant improves robustness or identity while introducing trade-offs in fidelity or faithfulness. We believe this extension will strengthen the practical relevance of our proposed framework by showing that MCDA is useful not only for comparing substantially different explainer but also for distinguishing between closely related explainer variants where manual comparison is less straightforward.
>
> In Table 4, we also included a balanced setting, in which all explanation evaluation metrics are considered together, alongside the single-metric-focused settings. The purpose of the single-metric settings was to reflect a user scenario in which one particular explanation property is prioritized over the others. In other words, our intention in the current version was to show how the rankings produced by SAW, TOPSIS, and ELECTRE I change when the decision process is driven by a single evaluation metric, while also providing a balanced comparison. We would also like to clarify that the main objective of this work was not to design mixed-preference settings that jointly prioritize combinations such as fidelity and stability. Rather, our goal was to show how disagreement among explainers can be resolved through MCDA when all explanation evaluation metrics are considered together, and additionally to illustrate how the ranking changes when a user has a specific preference for one metric. We will revise the manuscript to clarify the motivation and better explain the roles of both the balanced and single-metric-focused settings within the overall framework.
>
> Regarding the minor corrections, we will address all of them in the revised version of the manuscript.

---

### Decision · Action_Editor_rR3J · 2026-04-04

**Recommendation:** Reject

**Audience:**

Yes

**Audience Explanation:**

The paper tackles the important problem and it confirms that no single explainer dominates all metrics, however, the reviewers point out that the community interested in this topic could be very small.

**Claims And Evidence:**

No

**Claims Explanation:**

All the reviewers agree on this, I quote their comments by listing them below:
1) Two of the core evaluation metrics are either incorrectly specified or insufficiently justified.
(2) Limited experimental validation is done on explainers’ variants.
(3) There is a lack of empirical studies under non-trivial user preference settings. Please look at requested changes for details.
(4) The work does not demonstrate any downstream impact of the proposed approach, such as improved alignment with human judgments or improved predictive performance.
(5) No Comparison to explanation aggregation/ensemble methods [1],[2]
(6) Relatedly, the paper does not really show that users can set weights in a reliable, model- and dataset-aware way; nor does it evaluate whether different reasonable weight choices lead to stable selections, or whether the chosen explainer improves downstream outcomes (debugging, trust calibration, error detection, decision quality, etc.). So the evidence is clear enough for the mechanics of the framework, but not convincing for the central “resolution” claim and its practical implications.

**Resubmission Of Major Revision:**

The authors may consider submitting a major revision at a later time.